# ATTRIBUTE-BASED VISUAL REPROGRAMMING FOR VISION-LANGUAGE MODELS

**Chengyi Cai**[1]  **Zesheng Ye**[1]  **Lei Feng**[2,3]  **Jianzhong Qi**[1]  **Feng Liu**[1*]
[1]The University of Melbourne   [2]Southeast University   [3]Idealism Technology (Beijing)
{chengyi.cai1,zesheng.ye,jianzhong.qi}@unimelb.edu.au
lfengqaq@gmail.com   fengliu.ml@gmail.com

## ABSTRACT

*Visual reprogramming* (VR) reuses pre-trained vision models for downstream image classification tasks by adding trainable noise patterns to inputs. When applied to vision-language models (e.g., CLIP), existing VR approaches follow the same pipeline used in vision models (e.g., ResNet, ViT), where ground-truth class labels are inserted into fixed text templates to guide the optimization of VR patterns. This label-based approach, however, overlooks the rich information and diverse attribute-guided textual representations that CLIP can exploit, which may lead to the misclassification of samples. In this paper, we propose *Attribute-based Visual Reprogramming* (AttrVR) for CLIP, utilizing *descriptive attributes* (DesAttrs) and *distinctive attributes* (DistAttrs), which respectively represent common and unique feature descriptions for different classes. Besides, as images of the same class may reflect different attributes after VR, AttrVR iteratively refines patterns using the $k$-nearest DesAttrs and DistAttrs for each image sample, enabling more dynamic and sample-specific optimization. Theoretically, AttrVR is shown to reduce intra-class variance and increase inter-class separation. Empirically, it achieves superior performance in 12 downstream tasks for both ViT-based and ResNet-based CLIP. The success of AttrVR facilitates more effective integration of VR from unimodal vision models into vision-language models. Our code is available at https://github.com/tmlr-group/AttrVR.

## 1 INTRODUCTION

Recent studies (Xu et al., 2024; Chen et al., 2024; Wang et al., 2024) have demonstrated that downstream tasks can be efficiently addressed by repurposing pre-trained models from data-rich domains. For repurposing pre-trained image classifiers with a fixed label space (e.g., pre-trained ResNet (He et al., 2016), ViT (Dosovitskiy, 2020)), *visual reprogramming* (VR) (Cai et al., 2024b; Chen et al., 2023; Chen, 2024), also known as adversarial reprogramming (Tsai et al., 2020; Elsayed et al., 2019), is a model-agnostic technique that adjusts the input space while preserving the original models. VR (full problem setup detailed in Appendix A.1) trains additive noise patterns on images using downstream samples and their corresponding labels. Recently, VR has been extended to *vision-language models* (VLMs), such as CLIP (Radford et al., 2021), for downstream image classification. Existing implementations of VR for CLIP (Oh et al., 2023; Bahng et al., 2022) also follow the pipeline of vision models (i.e, image classifiers), relying on *template-prompted ground-truth labels* (e.g., 'This is a photo of [label]') to train the noise patterns.

However, VLMs are intrinsically different from unimodal vision classifiers in their capability to align attribute descriptions with image embeddings. Using label-based VR methods might fail to fully leverage such capability. Besides, similar syntactic structures in *template-prompted ground-truth labels* imply approximate text embeddings, leading to misclassifications of samples. Figure 1(a) shows the t-SNE (Van der Maaten & Hinton, 2008) embedding visualization results (the upper plot) of images with VR patterns (the lower plot) learned by template-prompted labels. Classes 'British Shorthair' and 'Russia Blue' from the OxfordPets (Parkhi et al., 2012) dataset are used as examples. Many samples are observed to have similar distances to the cluster centers of both classes,

---

*Correspondence to Feng Liu (fengliu.ml@gmail.com).

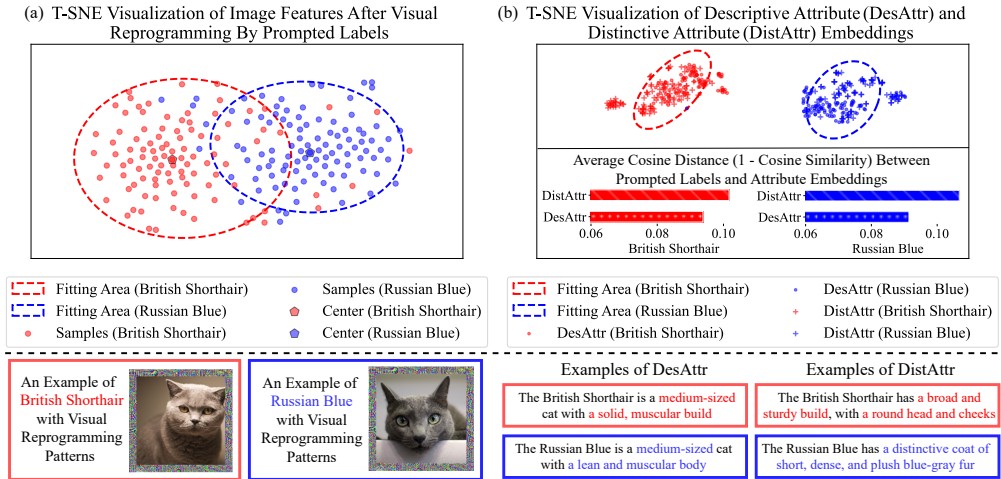

Figure 1: T-SNE visualization results of (a) embeddings of images with label-based (i.e., 'This is a photo of [label]') VR and (b) embeddings of text DesAttrs and DistAttrs for classes 'British Shorthair' and 'Russia Blue'. Examples of images with VR or attributes are shown below. Misclassifications occur in images with label-based VR, whereas attributes are easily distinguishable.

making them prone to misclassification. In contrast, Figure 1(b) shows the text embeddings visualization results of attributes generated by *large language model* (LLM) GPT-3.5 (Brown, 2020) given the class name 'British Shorthair' and 'Russia Blue', respectively. The text embeddings of attributes exhibit greater distinguishability compared to the embeddings of images with label-based approaches. Further, we use *Descriptive attributes* (DesAttrs) marked with '.' to denote the common characteristics of certain classes (with examples shown in the figure), and *distinctive attributes* (DistAttrs) marked with '+' to describe features that differentiate the class from others or exhibit individual differences. An important observation is that DesAttrs primarily concentrate near the cluster centers, while DistAttrs tend to be located away from the non-intended classes, e.g., DistAttrs of the red cluster are away from the blue cluster, especially the bottom left ones. The same is observed at the top-right corner of the blue cluster. This is also confirmed by the cosine distance: Eembeddings of DistAttrs are farther away from the prompted labels compared with DesAttrs (Figure 1(b)).

Such observations suggest that guiding VR training with DesAttrs and DistAttrs could improve classification accuracy compared with *template-prompted ground-truth labels*. In Section 4, we formalize DesAttrs and DistAttrs (Definitions 2 and 3) and propose *Attribute-based Visual Reprogramming* (AttrVR), which harnesses the attribute-querying ability of LLMs to describe DesAttrs and DistAttrs, capturing *multiple* common and unique features for each downstream class. Moreover, as images of the same class may reflect different attributes with evolving VR patterns, AttrVR queries the $k$-nearest DesAttrs and DistAttrs for individual image samples at each training epoch. By iteratively updating the VR patterns with *sample-specific attributes*, AttrVR fosters more context-aware image-attribute alignment and mitigates the ambiguity caused by *fixed template-prompted labels*.

In Section 5, we further establish that guiding the representation learning with DesAttrs and DistAttrs reduces intra-class variation and increases inter-class separation of image representations. This yields a more discriminative embedding space, thereby facilitating classification performance.

Experiments conducted on 12 widely-used benchmarks demonstrate the effectiveness of AttrVR in Section 6. AttrVR consistently outperforms other VR methods when using different encoder backbones or fewer training samples. Visualizations of the embedding space and individual samples with their top-matched attributes also substantiate the efficacy of AttrVR. Additional ablation, hyper-parameter (see Section 6) and aggregation studies (see Appendix C.3) further examine the contributions of different components within AttrVR.

Overall, both theoretical analysis and experimental results demonstrate that AttrVR has a clear advantage over label-based VR when applying CLIP to downstream classification tasks. The introduction of AttrVR represents a meaningful step towards adapting VR from repurposing single-modal pre-trained models with predefined label space to multimodal models (i.e., VLMs) for classification.

## 2 RELATED WORKS

**Prompting in Classification.** Prompt learning (Jia et al., 2022; Bahng et al., 2022; Oh et al., 2023) enables efficient adaptation of large pre-trained models to specific downstream tasks without fully finetuning the original models. Prompts can be trainable parameters integrated into different regions of pre-trained models. For vision models like ViT (Dosovitskiy, 2020), VPT (Jia et al., 2022) incorporates prompts in conjunction with the embeddings of input patches in each layer. EEVPT (Han et al., 2023) and TransHP (Wang et al., 2023) improve VPT by adding prompts within self-attention layers or learning prompt tokens for encoding coarse image categories.

For VLMs such as CLIP (Radford et al., 2021), prompt learning methods are typically based on few-shot samples from downstream tasks. CoOP (Zhou et al., 2022b) optimizes the text prompts focusing on the text encoder part of CLIP, while CoCoOP (Zhou et al., 2022a) further improves it by conditioning text prompts on input images. Beyond text prompts, MaPLe (Khattak et al., 2023) develops layer-specific mapping functions to connect visual and text prompts. PromptKD (Li et al., 2024b) appends a projection to the image encoder and employs knowledge distillation to train the prompts.

**Model Reprogramming and Input VR.** In contrast to prompting methods that introduce parameters within the model, model reprogramming (Chen, 2024) modifies the input and output spaces of downstream tasks. Therefore, it does not require meticulous design of parameter placement and is compatible with any model architecture. It has been applied in repurposing language (Hambardzumyan et al., 2021; Vinod et al., 2020), graph (Jing et al., 2023), vision (Tsai et al., 2020; Chen et al., 2023; Cai et al., 2024a) and acoustic models (Yang et al., 2021; 2023a; Hung et al., 2023).

Input VR refers to methods that add trainable noise patterns to images to repurpose pre-trained models, being model-agnostic and preserving the original model parameters. The differences between input VR, visual prompting and finetuning are outlined in Appendix A.1. Recent work on unimodal vision classifiers adds trainable noise that overlays resized images (Cai et al., 2024b) or pads around (Elsayed et al., 2019; Tsai et al., 2020; Chen et al., 2023) images, and then optimizes noise patterns using ground-truth labels. When applying VR to VLMs (Chen et al., 2023; Bahng et al., 2022; Oh et al., 2023), template-prompted ground-truth labels are used to train the noise patterns.

**Visual Attribute Query.** Visual Attribute Query (Pratt et al., 2023) refers to querying an LLM to obtain the corresponding visual features given downstream task labels. Current studies improve the zero-shot generalization performance (Pratt et al., 2023; Menon & Vondrick, 2023; Li et al., 2024a) and machine learning interpretability (Yang et al., 2023b; Yan et al., 2023) for VLMs.

Pratt et al. (2023) and Menon & Vondrick (2023) utilize GPT-3 (Brown, 2020) to generate descriptions of downstream task labels, thereby enhancing zero-shot classification accuracy. LaBo (Yang et al., 2023b) extends this approach by generating thousands of candidate concepts and constructing a class-concept weight matrix. To address the impact of redundancy in attribute descriptions, Yan et al. (2023) learn a concise set of attributes, Tian et al. (2024) introduce attribute sampling and proposes class-agnostic negative prompts, while WCA (Li et al., 2024a) calculates the similarity between descriptions and local visual regions.

## 3 PRELIMINARIES

**CLIP-based Classification.** CLIP (Radford et al., 2021) is a pre-trained VLM with an image encoder $f_{\text{img}} : \mathcal{X}^{\text{S}} \to \mathcal{Z}$ and a text encoder $f_{\text{txt}} : \mathcal{V} \to \mathcal{Z}$, where $\mathcal{X}^{\text{S}} \subseteq \mathbb{R}^{d_{\text{S}}}$ is a $d_{\text{S}}$-dimensional image space, and $\mathcal{V}$ is the text space. These encoders map an image $X^{\text{S}} \in \mathcal{X}^{\text{S}}$ and a text description $V \in \mathcal{V}$, into a shared embedding space $\mathcal{Z} \subseteq \mathbb{R}^d$. Then, the embedding similarity score between the image and the text description is calculated as

$$\text{sim}_{\text{CLIP}}(X^{\text{S}}, V) = \cos\left(Z_{\text{img}}, Z_{\text{txt}}\right)/\tau, \text{ with } Z_{\text{img}} = f_{\text{img}}(X^{\text{S}}), \text{ and } Z_{\text{txt}} = f_{\text{txt}}(V), \quad (1)$$

where $\cos(\cdot, \cdot)$ denotes cosine similarity and $\tau$ is a temperature parameter. Upon pre-training, CLIP can align semantically similar image-text pairs by maximizing their embedding similarity scores.

When it comes to a downstream classification task defined over $\mathcal{X}^{\text{T}} \times \mathcal{Y}^{\text{T}}$, where $\mathcal{X}^{\text{T}} \subseteq \mathbb{R}^{d_{\text{T}}}$ is a $d_{\text{T}}$-dimensional image space and $\mathcal{Y}^{\text{T}}$ is the label space of the downstream task, CLIP employs label prompting. For example, given the downstream label variable $Y^{\text{T}} \in \mathcal{Y}^{\text{T}}$, $\text{TP}(Y^{\text{T}}) \triangleq$

"This is a photo of" $\| Y^{\mathrm{T}}$, where $\|$ denotes concatenation, is commonly used to map a label $y^{\mathrm{T}} \in \mathcal{Y}^{\mathrm{T}}$ into a text description. Following, CLIP leverages the pre-trained visual-text alignment capability and assigns a label to $X^{\mathrm{T}}$ by selecting the most similar $\mathrm{TP}(Y^{\mathrm{T}} = y^{\mathrm{T}})$ in the embedding space $\mathcal{Z}$. Then, for a shape-compatible image $x^{\mathrm{T}}$, i.e., $d_{\mathrm{T}} = d_{\mathrm{S}}$, label prediction follows $\arg\max_{Y^{\mathrm{T}} \in \mathcal{Y}^{\mathrm{T}}} p_{\mathrm{CLIP}}(Y^{\mathrm{T}} \mid X^{\mathrm{T}})$ with a normalized conditional probability:

$$p_{\mathrm{CLIP}}(Y^{\mathrm{T}} = y^{\mathrm{T}} \mid X^{\mathrm{T}} = x^{\mathrm{T}}) = \frac{\exp\left(\mathrm{sim}_{\mathrm{CLIP}}(x^{\mathrm{T}}, \mathrm{TP}(y^{\mathrm{T}}))\right)}{\sum_{y' \in \mathcal{Y}^{\mathrm{T}}} \exp\left(\mathrm{sim}_{\mathrm{CLIP}}(x^{\mathrm{T}}, \mathrm{TP}(y'))\right)}. \tag{2}$$

**Input VR for CLIP-based Classification**. Input VR (Cai et al., 2024b) extends the applicability of frozen pre-trained models (e.g., CLIP) to downstream tasks with mismatched input shape, i.e., $d_{\mathrm{T}} \neq d_{\mathrm{S}}$. It introduces a learnable input transform $f_{\mathrm{in}} : \mathbb{R}^{d_{\mathrm{T}}} \to \mathbb{R}^{d_{\mathrm{S}}}$ defined as $f_{\mathrm{in}}(X^{\mathrm{T}}|\delta) \triangleq \mathrm{Pad}(X^{\mathrm{T}}) + \delta \odot M$, where $\mathrm{Pad}(\cdot)$ zero-pads around the input image and $\delta \in \mathbb{R}^{d_{\mathrm{S}}}$ are trainable parameters. $M$ is a binary mask with '0's in the area where $X^{\mathrm{T}}$ is located and '1's in the padding area. The Hadamard product $\odot$ ensures that $\delta$ only affects the padded regions. Thus, the transformed image is given by:

$$\tilde{X}^{\mathrm{T}} = f_{\mathrm{in}}(X^{\mathrm{T}}|\delta) \in \mathbb{R}^{d_{\mathrm{S}}}, \tag{3}$$

which allows CLIP to process inputs from $\mathcal{X}^{\mathrm{T}}$ by embedding them in $\mathcal{X}^{\mathrm{S}}$ with learned contextual information. With $\mathrm{TP}(Y^{\mathrm{T}})$ that maps class label to a text description, the VR-adapted CLIP prediction $p_{\mathrm{vr}}(Y^{\mathrm{T}}|X^{\mathrm{T}}) \propto \exp(\mathrm{sim}_{\mathrm{CLIP}}(\tilde{X}^{\mathrm{T}}, \mathrm{TP}(Y^{\mathrm{T}})|\delta))$ essentially follows Eq. (2) but is adapted for transformed input images. Given a downstream dataset $\mathcal{D}^{\mathrm{T}} = \{(x_i^{\mathrm{T}}, y_i^{\mathrm{T}})\}_{i=1}^N \overset{\mathrm{i.i.d}}{\sim} \mathcal{X}^{\mathrm{T}} \times \mathcal{Y}^{\mathrm{T}}$, where $N = n \times |\mathcal{Y}^{\mathrm{T}}|$ with $n$ samples per class, the optimization of VR pattern $\delta$ is driven by the cross-entropy loss, such that

$$\delta^* = \arg\min_{\delta} -\frac{1}{N} \sum_{i=1}^N \left[\log p_{\mathrm{vr}}(Y^{\mathrm{T}} = y_i^{\mathrm{T}}|X^{\mathrm{T}} = x_i^{\mathrm{T}})\right]. \tag{4}$$

**Limitation of** $\mathrm{TP}(Y^{\mathrm{T}})$. Eq. (4) implies that the optimization of $\delta$ exclusively relies on $\mathrm{TP}(Y^{\mathrm{T}})$ for supervision. However, as two labels $y_p, y_q \in \mathcal{Y}^{\mathrm{T}}$ share similar syntactic structures in their fixed template-based $\mathrm{TP}(Y^{\mathrm{T}} = y_p)$ and $\mathrm{TP}(Y^{\mathrm{T}} = y_q)$, the image-class embedding similarity scores, $\mathrm{sim}_{\mathrm{CLIP}}(\tilde{x}^{\mathrm{T}}, \mathrm{TP}(y_p))$ and $\mathrm{sim}_{\mathrm{CLIP}}(\tilde{x}^{\mathrm{T}}, \mathrm{TP}(y_q))$, may differ only slightly for the same input $x^{\mathrm{T}}$. This marginal gap in similarity scores heightens the misclassification risks, particularly in few-shot settings, where the small sample size exacerbates the challenge of resolving ambiguities between closely related text prompts. Figure 1(a) illustrates this concern, highlighting the potential classification errors due to the limited discriminative power of the text prompts.

## 4 ATTRIBUTE-BASED VISUAL REPROGRAMMING

**Describing Classes with Attributes**. The aforementioned limitation calls for more *informative* and *discriminative* information of label beyond $\mathrm{TP}(Y^{\mathrm{T}})$. Motivated by Figure 1(b), we leverage visual attributes (Ferrari & Zisserman, 2007) to capture more fine-grained visual features specific to each class than label-only representations. We thus propose attribute-based VR (AttrVR) that substitutes *template-prompted ground-truth labels* with *descriptive* and *distinctive* attributes, based on CLIP's pre-trained visual-text alignment capability. AttrVR aligns image representations with fine-grained attributes-based representations of classes that more effectively distinguish different classes than template-prompted labels. We begin by formalizing relevant concepts used in AttrVR.

**Definition 1** (Attributes). *Let $\mathcal{X}$ be the input space of images, $\mathcal{Y}$ be the set of class labels, and $\mathcal{A}$ be the universal set of all possible attributes (e.g., tall plant, red color). Define a mapping $f_m$ from a class label $y$ to a set of attributes $\mathcal{A}(y)$. For each attribute $a \in \mathcal{A}$, define an indicator function $f_a : \mathcal{X} \to \{0,1\}$, such that $f_a(x) = 1$ if attribute $a$ is identified in input $x$ based on a specified similarity criterion[1], and 0 otherwise. Then, for any class $y \in \mathcal{Y}$, the set of attributes $\mathcal{A}(y)$ is connected to images by the features that characterize samples belonging to that class.*

To further characterize the attributes most relevant for class description and distinction, we introduce two subsets from $\mathcal{A}(y)$, namely *descriptive attributes* (DesAttrs) and *distinctive attributes* (DistAttrs). DesAttrs refer to the *most common* visual features across multiple samples belonging to the same class, describing the class by capturing its general characteristics.

---

[1]The criterion can vary based on different contexts (Kumar et al., 2011; Pham et al., 2021). In this study, we focus on CLIP similarity in the embedding space $\mathcal{Z}$ induced by VLM, which will be elaborated on in Section 4.

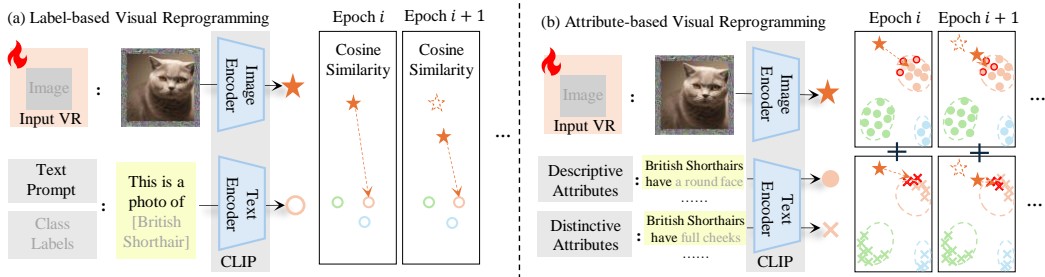

Figure 2: The comparison of (a) previous label-based VR and (b) our attribute-based VR. Previous VR methods use fixed template-prompted ground-truth labels for all samples to optimize the VR pattern $\delta$ (using Eq. (2) and Eq. (3)), whereas our method iteratively selects $k$ nearest DesAttrs and DistAttrs for individual samples in each epoch to optimize the VR pattern $\delta$ (using Eq. (9)).

**Definition 2** (Descriptive Attributes). *For a class $y \in \mathcal{Y}$ and a set cardinality $m \in \mathbb{N}_+$, DesAttrs are defined as the $m$ most frequently identified attributes from the samples within class $y$,*

$$\mathcal{A}_{\text{des}}(y) \triangleq \{a_i \in \mathcal{A}(y) \mid U_y(a_i) \geq U_y(a), \forall i \in \{1, \ldots, m\}, \forall a \in \mathcal{A}(y) \setminus \{a_1, \ldots, a_i\}\}, \quad (5)$$

*where $\mathcal{X}_y$ denotes the set of all images in class $y$, and $U_y(a) = \sum_{x \in \mathcal{X}_y} f_a(x)/|\mathcal{X}_y|$ is the frequency of attribute $a \in \mathcal{A}$ in class $y \in \mathcal{Y}$. $U_y(a)$ can be used to rank the $m$ highest-frequency attributes.*

In contrast, DistAttrs are visual features that distinguish a class from other classes, appearing in the class while being the *least common* in the other classes.

**Definition 3** (Distinctive Attributes). *For a class $y \in \mathcal{Y}$ and a set cardinality $m \in \mathbb{N}_+$, DistAttrs are defined as the $m$ attributes that are most uniquely associated with class $y$,*

$$\mathcal{A}_{\text{dist}}(y) \triangleq \{a_i \in \mathcal{A}(y) \mid V_y(a_i) \geq V_y(a), \forall i \in \{1, \ldots, m\}, \forall a \in \mathcal{A}(y) \setminus \{a_1, \ldots, a_i\}\}, \quad (6)$$

*where $V_y(a) = 1 - (\sum_{y' \in \mathcal{Y} \setminus \{y\}} \sum_{x \in \mathcal{X}_{y'}} f_a(x)/(|\mathcal{X}| - |\mathcal{X}_y|))$ calculates the presence of an attribute $a \in \mathcal{A}$ in samples of class $y \in \mathcal{Y}$ against its presence in all other classes $y' \in \mathcal{Y} \setminus \{y\}$.*

Intuitively, describing $\mathcal{A}_{\text{des}}(y) \cup \mathcal{A}_{\text{dist}}(y)$ leads to more information of $y$ than relying on $\text{TP}(y)$.

**Method Overview.** For downstream image classification, AttrVR follows the general input VR pipeline by optimizing the padded VR noise pattern $\delta$ over a dataset $\mathcal{D}^{\text{T}} = \{(x_i^{\text{T}}, y_i^{\text{T}})\}_{i=1}^N$ drawn from $\mathcal{X}^{\text{T}} \times \mathcal{Y}^{\text{T}}$, as introduced in Section 3. Yet, it diverges from label-based VR approaches in two key strategies (see Figure 2). First, for each label $y^{\text{T}} \in \mathcal{Y}^{\text{T}}$, AttrVR replaces previously used text prompts $\text{TP}(Y^{\text{T}} = y^{\text{T}})$, adopts DesAttrs (Definition 2) and DistAttrs (Definition 3) that describe common and unique attributes of $y^{\text{T}}$ as the supervision signal. Second, AttrVR employs a $k$-nearest neighbor iterative updating strategy to ensure that attribute assignments are continuously refined, allowing the most relevant attributes for each sample to adapt dynamically as the trainable noise $\delta$ evolves across epochs. The detailed strategies are elaborated on below.

**Generating Attributes with LLMs.** The concept of attributes (Definition 1) is built upon $f_m$ that maps class labels to subsets of $\mathcal{A}$. However, $f_m$ is intractable due to the exponential growth of the possible number of attributes, making direct computation and storage of all possible attribute combinations impractical. Moreover, manually defining attributes for each class is also infeasible in complex domains where attributes may not be easily enumerated or predefined. To this end, we use powerful LLMs with *visual attribute query* capabilities (Pratt et al., 2023), denoted by $f_{\text{LLM}}(Y^{\text{T}})$, as a tractable surrogate for implementing $f_m(Y^{\text{T}})$. LLMs can infer relevant attributes for any class with context-driven queries, bypassing the need to compute the entire power set of $\mathcal{A}$ – they generate $\mathcal{A}_{\text{des}}(Y^{\text{T}} = y^{\text{T}})$ and $\mathcal{A}_{\text{dist}}(Y^{\text{T}} = y^{\text{T}})$ according to different downstream tasks and class labels.

Concretely, we adopt GPT-3.5 (Brown, 2020) to generate $\mathcal{A}_{\text{des}}(y^{\text{T}})$ and $\mathcal{A}_{\text{dist}}(y^{\text{T}})$ each containing $m$ attributes, by prompting the LLM with task-specific and class-specific queries, formulated as

$$\tilde{\mathcal{A}}_{\text{des}}(y^{\text{T}}) = f_{\text{LLM}}(y^{\text{T}} | [\text{des\_prompt}]), \tilde{\mathcal{A}}_{\text{dist}}(y^{\text{T}}) = f_{\text{LLM}}(y^{\text{T}} | [\text{dist\_prompt}]). \quad (7)$$

As a result, we collect $2m$ attributes for each class $y^{\text{T}} \in \mathcal{Y}^{\text{T}}$, which will be used for optimizing $\delta$. The details of attribute generation (prompts, settings, etc.) are in Appendix A.2.1.

$k$-**nearest Iterative Updating Strategy.** Recall that CLIP-based image classification is performed upon ranking the image-text embedding similarity scores. However, the most similar attribute descriptions even from the same attribute set may vary between: (1) different images of the same class, i.e., inconsistencies of visual features among different samples, and (2) the same image with evolving VR patterns, i.e., changes in $\delta$ during training, leading to potential misalignment between image and relevant attributes. In response, we propose $k$-*nearest neighbor* attribute query to reduce the sensitivity to individual attributes for addressing (1) and employ an *iterative updating strategy* to adapt to changing VR patterns as a workaround for (2).

Specifically, consider the training dataset $\mathcal{D}^{\mathrm{T}}$ of the downstream task. For each downstream image $x_i^{\mathrm{T}}$, we first obtain its transformation $\tilde{x}_i^{\mathrm{T}}$ (*cf.* Eq. (3)). Then, we identify sample-specific $k$-nearest DesAttrs for $x_i^{\mathrm{T}}$ by computing the CLIP embedding similarity between $\tilde{x}_i^{\mathrm{T}}$ and all attributes from the LLM-generated DesAttrs $\tilde{\mathcal{A}}_{\mathrm{des}}(y^{\mathrm{T}})$, ranked in descending order of similarity, such that

$$\tilde{\mathcal{A}}_{\mathrm{des}}^k(x_i^{\mathrm{T}}, y^{\mathrm{T}}|\delta^{(e)}) = \{a_j\}_{j=1}^k : \mathrm{sim}_{\mathrm{CLIP}}(x_i^T, a_j|\delta^{(e)}) > \mathrm{sim}_{\mathrm{CLIP}}(x_i^T, a|\delta^{(e)}),$$
$$\forall a \in \tilde{\mathcal{A}}_{\mathrm{des}}(y^{\mathrm{T}}) \setminus \{a_1, \ldots, a_{j-1}\}. \tag{8}$$

Here, $\delta^{(e)}$ refers to the VR pattern in the training epoch $e$. Similarly, the sample-specific $k$-nearest DistAttrs $\tilde{\mathcal{A}}_{\mathrm{dist}}^k(x_i^{\mathrm{T}}, y^{\mathrm{T}}|\delta^{(e)})$ can be obtained in the same manner.

Then, the attribute-based embedding similarity score between $x_i^{\mathrm{T}}$ and $\forall y^{\mathrm{T}} \in \mathcal{Y}^{\mathrm{T}}$, which incorporates its both sample-specific $\tilde{\mathcal{A}}_{\mathrm{des}}^k \triangleq \tilde{\mathcal{A}}_{\mathrm{des}}^k(x_i^{\mathrm{T}}, y^{\mathrm{T}}|\delta^{(e)})$ and $\tilde{\mathcal{A}}_{\mathrm{dist}}^k \triangleq \tilde{\mathcal{A}}_{\mathrm{dist}}^k(x_i^{\mathrm{T}}, y^{\mathrm{T}}|\delta^{(e)})$, is computed by a weighted aggregation:

$$\mathrm{sim}_{\mathrm{Attr}}(x_i^{\mathrm{T}}, y^{\mathrm{T}}|\delta^{(e)}) = \frac{\lambda}{k} \sum_{a \in \tilde{\mathcal{A}}_{\mathrm{des}}^k} \mathrm{sim}_{\mathrm{CLIP}}(\tilde{x}_i^{\mathrm{T}}, a|\delta^{(e)}) + \frac{1-\lambda}{k} \sum_{a' \in \tilde{\mathcal{A}}_{\mathrm{dist}}^k} \mathrm{sim}_{\mathrm{CLIP}}(\tilde{x}_i^{\mathrm{T}}, a'|\delta^{(e)}), \tag{9}$$

where $\lambda \in [0, 1]$ balances the contribution of DesAttrs and DistAttrs. Then, the predictive probability $p_{\mathrm{vr}}(Y^{\mathrm{T}} = y_i^{\mathrm{T}}|X^{\mathrm{T}} = x_i^{\mathrm{T}})$ is determined for each sample $x_i^{\mathrm{T}}$ with a $\mathrm{softmax}(\cdot)$ resembling Eq. (2), but now with a new attribute-based embedding similarity score $\mathrm{sim}_{\mathrm{Attr}}(x_i^{\mathrm{T}}, y^{\mathrm{T}}|\delta^{(e)})$ evaluated at each epoch $e$. We iteratively update the VR pattern, optimizing parameters $\delta^{(e+1)} \leftarrow \delta^{(e)} - \alpha \nabla_\delta^{(e)}$ with respect to the cross-entropy loss (*cf.* Eq. (4)) under learning rate $\alpha$, over the training dataset $\mathcal{D}^{\mathrm{T}}$.

---

**Algorithm 1** Training Pipeline of AttrVR

---

1: **Input:** Few-shot training data $\mathcal{D}^{\mathrm{T}} = \{(x_i^{\mathrm{T}}, y_i^{\mathrm{T}})\}_{i=1}^N$, hyper-parameters $k, \lambda$, learning rate $\alpha$, epoch number $E$, and pre-trained CLIP model
2: **Output:** Trained VR pattern $\delta^{(E)}$ applying AttrVR
3: # Step 1: Calculate and Store Attribute Embeddings
4: **for** $y \in \mathcal{Y}^{\mathrm{T}}$ **do**
5:     Obtain $\tilde{\mathcal{A}}_{\mathrm{des}}(y)$ and $\tilde{\mathcal{A}}_{\mathrm{dist}}(y)$ by Eq. (7)
6:     Get $Z_{\mathrm{txt}}(a)$ for $\forall a \in \tilde{\mathcal{A}}_{\mathrm{des}}(y) \cup \tilde{\mathcal{A}}_{\mathrm{dist}}(y)$
7: **end for**
8: # Step 2: Begin Training the VR Pattern
9: Initialize $\delta^{(0)} \leftarrow \{0\}^{d_{\mathrm{S}}}$
10: **for** $e = 0$ **to** $E - 1$ **do**
11:     **for** $i = 1$ **to** $N$ **do**
12:         Compute $\tilde{\mathcal{A}}_{\mathrm{des}}^k(x_i^{\mathrm{T}}, y|\delta^{(e)}), \tilde{\mathcal{A}}_{\mathrm{dist}}^k(x_i^{\mathrm{T}}, y|\delta^{(e)})$ by Eq. (8) with Stored Embeddings for $\forall y \in \mathcal{Y}^{\mathrm{T}}$
13:         Compute $p_{\mathrm{vr}}(y_i^{\mathrm{T}}|x_i^{\mathrm{T}})$ by Eq. (9)
14:     **end for**
15:     $\delta^{(e+1)} \leftarrow \delta^{(e)} - \alpha \nabla_\delta^{(e)}$ # Iterative Update
16: **end for**

---

**Comparison with Label-based VR**. Besides using easily distinguishable attributes that replace previous template-prompted labels with similar syntactic structures to facilitate classification, AttrVR also aligns with the evolving nature of $\delta$. In contrast to label-based VR that aligns images of the same class with a *fixed* $\mathrm{TP}(y^{\mathrm{T}})$, AttrVR re-queries $\tilde{\mathcal{A}}_{\mathrm{des}}^k(x_i^T, y^{\mathrm{T}}|\delta^{(e)})$ and $\tilde{\mathcal{A}}_{\mathrm{dist}}^k(x_i^T, y^{\mathrm{T}}|\delta^{(e)})$ for *each image* at *every epoch*. This enables AttrVR to iteratively refine image-attribute alignment, yielding refined $\mathrm{sim}_{\mathrm{Attr}}(x_i^T, y^{\mathrm{T}}|\delta^{(e)})$ over epochs. In other words, while both label-based VR and AttrVR target the cross-entropy objective, AttrVR benefits from contextually relevant optimization with *sample-specific* $k$-nearest attributes as supervision signals.

**Training Pipeline and Efficiency.** Algorithm 1 outlines the pipeline of AttrVR. We note that the text embeddings of DesAttrs and DistAttrs are *pre-computed before training*, introducing negligible computational overhead compared to label-based VR. See Appendix C.5 for details.

## 5    UNDERSTANDING THE EFFECTS OF ATTRIBUTES

This section will justify why DesAttrs and DistAttrs would facilitate classification. The ease of classification decision boundary depends on class separability (Lorena et al., 2019), which quantifies how well different classes can be distinguished in the embedding space. This measure is jointly determined by *intra-class variance* (i.e., the spread of embeddings within a class) and *inter-class distance* (i.e., the separation of embeddings from different classes).

**Definition 4** (Class Separability). *Let $\mathcal{X}$ and $\mathcal{Y}$ be the input and class spaces as in Definition 1. Let $\mathcal{Z}$ be the image embedding space induced by the image encoder $Z_{\text{img}} : \mathcal{X} \to \mathcal{Z}$. For each class $y \in \mathcal{Y}$, let $\mathcal{X}_y$ be the set of all images with label $y$, and let $\mu_y = \sum_{x \in \mathcal{X}_y} z_{\text{img}}(x)/|\mathcal{X}_y|$ be the mean image embedding of class $y$. Then, class separability (CS) is defined as:*

$$CS(\mathcal{Y}; \mathcal{Z}) = -\frac{1}{|\mathcal{Y}|} \sum_{y \in \mathcal{Y}} \underbrace{\frac{1}{|\mathcal{X}_y|} \sum_{x \in \mathcal{X}_y} \|Z_{\text{img}}(x) - \mu_y\|^2}_{\text{Tr}(\sigma^2(y)) \triangleq \textit{intra-class variation}} + \frac{1}{|\mathcal{Y}|(|\mathcal{Y}| - 1)} \sum_{y \neq y'} \underbrace{\|\mu_y - \mu_{y'}\|^2}_{d(y,y') \triangleq \textit{inter-class distance}} \quad,$$

The value of $CS(\mathcal{Y}; \mathcal{Z})$ measures the difference of average *intra-class variance*, i.e., $\text{Tr}(\sigma^2(y))$, and *inter-class distance*, i.e., $d(y, y')$, across all classes. A higher value indicates the image embeddings are better separated in the embedding space. Thus, the goal of maximizing class separability is equivalent to reducing *intra-class variation* while increasing *inter-class distance*.

**Lemma 1.** *Let $\mathcal{A}_{\text{des}}(y) \subseteq \mathcal{A}(y)$ be the set of descriptive attributes for class $y$ as with Definition 2. Let $\Sigma_{\text{A}}$ and $\Sigma_{\text{L}}$ be the covariance matrices of the embeddings optimized with respect to $\mathcal{A}_{\text{des}}(y)$ and $y$, respectively. Then, for any class $y \in \mathcal{Y}$, we have $\text{Tr}(\Sigma_{\text{A}}(y)) \leq \text{Tr}(\Sigma_{\text{L}}(y))$.*

Lemma 1 (details in Appendix B) shows that $\mathcal{A}_{\text{des}}(y)$ leads to reduced intra-class variances of image embeddings $Z_{\text{img}}(x)$, as the most frequently identified attributes in $\mathcal{X}_y$ imply that text embedding $Z_{\text{txt}}(a)$ of attributes closely align with $Z_{\text{img}}(x)$ for $x \in \mathcal{X}_y$. In addition, aggregating over $\mathcal{A}_{\text{des}}(y)$ pulls $Z_{\text{img}}(x)$ towards to class mean $\mu_y$, reducing the dispersion of per-class sample embeddings.

**Lemma 2.** *Let $\mathcal{A}_{\text{dist}}(y) \subseteq \mathcal{A}(y)$ be the set of distinctive attributes for class $y$ as with Definition 3. Let $d_{\text{A}}(y, y')$ and $d_{\text{L}}(y, y')$ be $\ell^2$ distance between mean embeddings of two classes $y \neq y'$, optimized with respect to $\mathcal{A}_{\text{dist}}(y)$ and $y$. Then, for any $y, y' \in \mathcal{Y}$, we have $d_{\text{A}}(y, y') \geq d_{\text{L}}(y, y')$ if $|\mathcal{A}_{\text{dist}}(y)| > |\mathcal{Y}|$, which is easy to satisfy since $|\mathcal{Y}|$ is fixed while the size of $\mathcal{A}_{\text{dist}}(y)$ is unrestricted.*

Lemma 2 (details in Appendix B) implies that $\mathcal{A}_{\text{dist}}(y)$ promotes inter-class separation. $\mathcal{A}_{\text{dist}}(y)$ is uniquely associated with class $y$ and minimally present in classes $y'$. For samples $x' \in \mathcal{X}_{y'}$, the similarity between $Z_{\text{img}}(x')$ and $Z_{\text{txt}}(a)$ is low for $a \in \mathcal{A}_{\text{dist}}(y)$. Thus, the mean embeddings of different classes are pushed further apart due to the minimal overlap between $\mathcal{A}_{\text{dist}}(y)$ and $\mathcal{A}_{\text{dist}}(y')$.

**Corollary 1.** *Let $\mathcal{Z}_{\text{A}}$ and $\mathcal{Z}_{\text{L}}$ be the embedding spaces obtained through attribute-based and label-based optimization. Denote the respective class separability by $CS(\mathcal{Y}; \mathcal{Z}_{\text{A}})$ and $CS(\mathcal{Y}; \mathcal{Z}_{\text{L}})$, as with Definition 4. Then, under the conditions of Lemmas 1 and 2, it holds that $CS(\mathcal{Y}; \mathcal{Z}_{\text{A}}) > CS(\mathcal{Y}; \mathcal{Z}_{\text{L}})$.*

Merits of attribute-based optimization inspire a practical VR solution. However, Lemmas 1 and 2 examine the effects of DesAttrs and DistAttrs in isolation, but attribute sets may overlap. Quantifying their combined effect is challenging due to the complex non-linearity of neural network optimization, making a careful balance between DesAttrs and DistAttrs essential for better performance.

## 6    EXPERIMENTS

**Baselines and Benchmarks.** To evaluate AttrVR, we use CLIP as the pre-trained model and conduct experiments on 12 downstream classification tasks with 16 shots for each class following Oh et al. (2023). These datasets encompass diverse visual domains, involving scenes, actions, textures, and fine-grained details (see Appendix A.2.2). We include four baselines, including (1) *ZS*, which is the zero-shot performance of CLIP, (2) *AttrZS*, which applies our DesAttrs and DistAttrs for zero-shot classification (see Appendix A.3), and state-of-the-art VR methods for VLMs: (3) *VP* (Bahng et al., 2022), which overlays VR patterns on resized images, and (4) *AR* (Tsai et al., 2020; Chen et al., 2023), which pads VR patterns around images. See the implementation details in Appendix A.2.3. Regarding hyper-parameters in AttrVR, we set $k = 3$ and $\lambda = 0.5$ and will discuss their impact.

Table 1: Accuracy comparison of different methods trained on 16-shot downstream classification tasks, using ViT-B16-based CLIP as the pre-trained model (Mean % ± Std %, ours are highlighted and the highest is in **bold**).

| Method | Aircraft | Caltech | Cars | DTD | ESAT | Flowers | Food | Pets | SUN | UCF | IN | Resisc | Avg. |
|---|---|---|---|---|---|---|---|---|---|---|---|---|---|
| ZS | 22.4 | 89.0 | 65.2 | 41.1 | 38.7 | 65.5 | 84.4 | 86.1 | 61.7 | 66.7 | 64.2 | 55.9 | 61.7 |
| AttrZS | 28.5 | 94.1 | 65.1 | 54.3 | 50.8 | 81.6 | **86.5** | 91.6 | 65.6 | 69.3 | 69.3 | 62.2 | 68.2 |
| VP | 32.1 | 93.5 | 65.5 | 61.4 | 91.2 | 82.5 | 82.3 | 91.0 | 65.8 | 73.8 | 64.2 | 79.1 | 73.5 |
|  | ±0.6 | ±0.1 | ±0.3 | ±0.5 | ±0.3 | ±0.4 | ±0.1 | ±0.3 | ±0.2 | ±0.5 | ±0.1 | ±0.3 |  |
| AR | 31.7 | 95.5 | 68.0 | 62.0 | 93.4 | 85.9 | 85.2 | 92.7 | 67.9 | 78.1 | 66.0 | 81.6 | 75.7 |
|  | ±0.3 | ±0.2 | ±0.3 | ±0.1 | ±0.1 | ±0.7 | ±0.1 | ±0.1 | ±0.3 | ±0.2 | ±0.0 | ±0.3 |  |
| AttrVR | **36.6** | **95.7** | **68.3** | **65.6** | **93.8** | **92.9** | 85.9 | **93.3** | **69.6** | **79.0** | **69.4** | **82.6** | **77.7** |
|  | ±0.3 | ±0.1 | ±0.3 | ±0.8 | ±0.3 | ±0.4 | ±0.1 | ±0.0 | ±0.1 | ±0.6 | ±0.0 | ±0.4 |  |

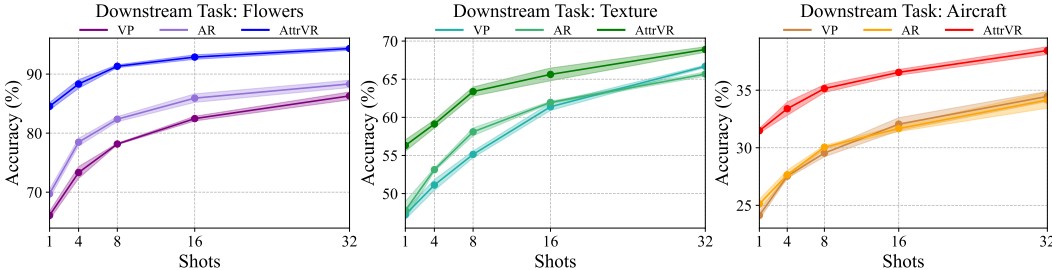

Figure 3: Accuracy comparison of different VR methods trained on different shots from [1, 4, 8, 16, 32]. Pre-trained ViT-B16-based CLIP is used. The striped area indicates the error bars.

**Overall Performance Comparison.** Using CLIP with a ViT-B16 visual encoder as the pre-trained model, the comparison results are shown in Table 1. It can be observed that even only using DesAttrs and DistAttrs for zero-shot classification already outperforms some baseline few-shot VR methods on the Caltech, Food, and SUN datasets. This demonstrates the effectiveness of DesAttrs and DistAttrs. However, VR methods remain necessary for datasets with significant domain differences, such as EuroSAT and DTD. AttrVR surpasses the baseline VR methods VP and AR across all datasets, achieving an average improvement of 2% over the state-of-the-art methods across the 12 datasets. The advantages of AttrVR are particularly notable in fine-grained classification tasks with distinct visual feature differences, such as Flowers (+7.0%), DTD (+3.6%), and Aircraft (+4.9%). On the Food dataset, AttrVR shows slightly lower accuracy than AttrZS, which may be because the images used by VR methods have a smaller size than those used in the zero-shot settings.

**Results of Sample Efficiency.** We evaluate the performances of all VR methods across sparse (1-, 4-, 8-shots) and abundant (32-shots) training settings on Flowers, Texture, and Aircraft datasets. Figure 3 shows that AttrVR maintains stronger performance across all settings than baselines, demonstrating both resilience to data scarcity and effective use of additional samples when available.

**Results on Different Backbones.** We investigate how VR methods perform across multiple pre-trained backbones from the CLIP model family, ranging from ResNet50 to ViT-L14. Table 2 presents the mean accuracy over 12 datasets, demonstrating that (1) all VR methods become more effective with more powerful backbones, and (2)AttrVR consistently outperforms baseline VR methods regardless of the backbone architectural scales. Detailed results and analysis for each dataset are provided in the Appendix C.1.

Table 2: Average accuracy of different VR methods on 12 datasets, using different backbones as CLIP visual encoders (Mean Accuracy %, ours are highlighted and the highest is in **bold**, RN stands for ResNet).

|  | RN50 | RN101 | ViT-B32 | ViT-B16 | ViT-L14 |
|---|---|---|---|---|---|
| ZS | 53.4 | 56.1 | 58.2 | 61.7 | 68.7 |
| AttrZS | 59.9 | 62.4 | 63.8 | 68.2 | 73.2 |
| VP | 53.2 | 57.1 | 67.5 | 73.5 | 61.1 |
| AR | 59.9 | 62.3 | 65.5 | 75.7 | 71.9 |
| AttrVR | **64.2** | **66.8** | **69.1** | **77.7** | **75.5** |

**Visualization Results of AttrVR.** Figure 4 illustrates the results of applying the trained VR pattern to images from the Flowers task with the label 'globe thistle' and images from the Texture task with the label 'Banded'. It also shows the closest DesAttrs and DistAttrs corresponding to these results. For 'globe thistle', the closest DesAttr primarily describes its height and width, while the DistAttr

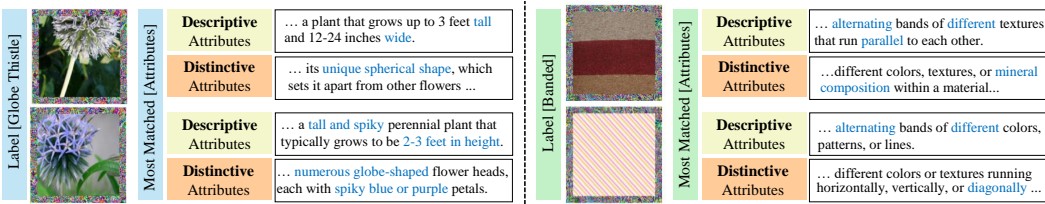

Figure 4: Visualization of images with AttrVR patterns, and their nearest DesAttrs and DistAttrs, using the ViT-B16-based CLIP as the pre-trained model. Two images labeled 'Globe Thistle' from 'Flowers' and two labeled 'Banded' from 'Texture' are chosen as examples (more in Appendix C.2).

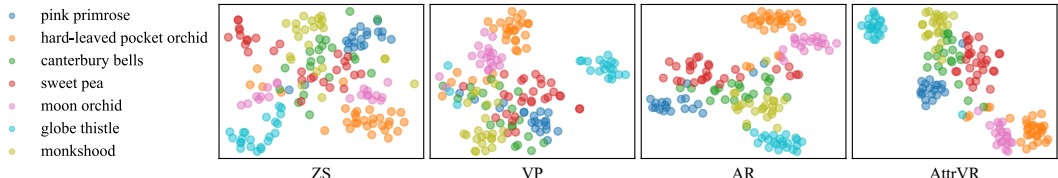

Figure 5: T-SNE visualization results of image embeddings from seven classes in the Flowers task, utilizing the ViT-B16-based CLIP as the pre-trained model. In the first plot, embeddings of zero-shot images are indicated with *ZS*. The following three plots display embeddings of images with VR patterns, categorized by different training methods and marked as *VP*, *AR*, and *AttrVR*, respectively.

highlights features that differentiate it from other flowers, such as its spherical shape. Different samples of 'globe thistle' may have different closest DistAttrs; for instance, the image with blue-violet petals will be closest to a DistAttr with a similar description. Similarly, for images with the 'Banded' label, the DesAttrs mainly describes the common feature of alternating textures, while the DistAttrs capture unique characteristics of the class or individuals, such as 'mineral composition' or 'diagonal textures' shown in Figure 4.

**Visualization Results of Embedding Space.** Figure 5 plots the 2D t-SNE embeddings (Van der Maaten & Hinton, 2008) of classifying samples from the Flowers task under input VR methods, with different colors representing different categories. It can be observed that in the zero-shot (*ZS*) scenario, some classes, such as 'moon orchid' (marked with pink dots), are scattered and difficult to classify. However, label-based VR methods, such as VP and AR, help to clarify the boundaries of these classes, making the samples easier to distinguish. Despite this improvement, some classes, like 'canterbury bells' (marked with green dots) and 'sweet pea' (marked with red dots) still remain relatively indistinguishable. After applying our AttrVR, the embeddings of various categories cluster more distinctly in the 2D visualization plane, resulting in clearer and more separable distributions.

**Ablation Studies.** Table 3 presents the ablation studies, and sequentially details: (1) *w/o VR*: the results of AttrVR without training the input VR, where only zero-padded images from the downstream task are classified by our DesAttrs and DistAttrs, along with k-nearest neighbor attribute selection for zero-shot results; (2) *w/o DesAttrs*: the results of training AttrVR utilizing only DistAttrs, excluding DesAttrs; (3) *w/o DistAttrs*: the results of training AttrVR utilizing only DesAttrs, excluding DistAttrs; (4) *w/o both Attrs*: the results without both attributes, which correspond to the results of the label-based VR approach; and (5) *Ours*: the results using our proposed method, AttrVR.

In the absence of training VR patterns, performance on downstream tasks can be unsatisfactory when there is a significant domain shift from the pre-trained CLIP model's domain. For instance, low accuracy is observed when the downstream tasks involve remote sensing datasets such as ESAT or Resisc, or texture datasets like DTD. Thus, training VR patterns is crucial for effectively adapting the pre-trained model to unfamiliar domains.

In the absence of DesAttrs, the method relies only on unique attributes during training, emphasizing class differences in the downstream task. This approach works well when the downstream domain closely matches the pre-trained model, as in broad classification tasks like Caltech. However, for tasks with significant domain shifts and few classes, such as the 10-class remote sensing dataset ESAT, it may miss some overall attributes relevant to certain classes, resulting in lower performance.

Table 3: Ablation studies of AttrVR, using ViT-B16-based CLIP as the pre-trained model (Mean % ± Std %, ours are  highlighted  and the highest is in **bold**).

| Method | Aircraft | Caltech | Cars | DTD | ESAT | Flowers | Food | Pets | SUN | UCF | IN | Resisc | Avg. |
|---|---|---|---|---|---|---|---|---|---|---|---|---|---|
| w/o VR | 25.4 | 94.1 | 62.3 | 54.3 | 48.5 | 80.8 | 84.8 | 91.6 | 64.3 | 68.4 | 68.0 | 61.2 | 67.0 |
| | ±0.4 | ±0.1 | ±0.1 | ±0.1 | ±0.3 | ±0.3 | ±0.1 | ±0.1 | ±0.0 | ±0.3 | ±0.1 | ±0.1 | |
| w/o DesAttrs | 36.1 | **95.9** | 68.2 | 64.8 | 93.1 | 92.6 | 85.8 | 93.3 | 69.4 | 78.0 | 69.3 | 81.9 | 77.4 |
| | ±0.4 | ±0.1 | ±0.1 | ±0.4 | ±0.2 | ±0.6 | ±0.0 | ±0.1 | ±0.0 | ±0.4 | ±0.1 | ±0.6 | |
| w/o DistAttrs | 35.9 | 95.6 | 68.2 | 64.4 | 93.8 | 92.4 | 85.7 | 93.0 | 67.7 | 78.6 | 68.9 | 81.8 | 77.2 |
| | ±0.3 | ±0.1 | ±0.2 | ±1.1 | ±0.3 | ±0.1 | ±0.0 | ±0.2 | ±0.2 | ±0.5 | ±0.0 | ±0.1 | |
| w/o both Attrs | 31.7 | 95.5 | 68.0 | 62.0 | 93.4 | 85.9 | 85.2 | 92.7 | 67.9 | 78.1 | 66.0 | 81.6 | 75.7 |
| | ±0.3 | ±0.2 | ±0.3 | ±0.1 | ±0.1 | ±0.7 | ±0.1 | ±0.1 | ±0.3 | ±0.2 | ±0.0 | ±0.3 | |
| Ours | **36.6** | 95.7 | **68.3** | **65.6** | **93.8** | **92.9** | **85.9** | **93.3** | **69.6** | **79.0** | **69.4** | **82.6** | **77.7** |
| | ±0.3 | ±0.1 | ±0.3 | ±0.8 | ±0.3 | ±0.4 | ±0.1 | ±0.0 | ±0.1 | ±0.6 | ±0.0 | ±0.4 | |

Figure 6: Performance comparison applying different hyper-parameters. The first row shows the impact weight $\lambda$ that balances DesAttrs and DistAttrs. The second row shows the impact of $k$, being the number of nearest attributes selected for classification. Pre-trained ViT-B16-based CLIP is used.

In the absence of DistAttrs, the method may overlook some attributes crucial for differentiating between categories in the downstream task. For datasets with many hard-to-differentiate classes, such as action classification datasets like UCF or texture classification datasets like DTD, not using DistAttrs can negatively impact classification performance. Besides, without both attributes, AttrVR degenerates into label-based VR, forfeiting its advantages in fine-grained classification tasks.

**Hyper-parameter Analyses.** Figure 6 illustrates the impact of the hyper-parameters $\lambda$ and $k$. The weight $\lambda$ is used to balance the contributions of DesAttrs and DistAttrs. For different tasks, the optimal $\lambda$ varies, with accuracy generally rising and then dropping as $\lambda$ increases, indicating a moderate $\lambda$ is needed to balance DesAttrs and DistAttrs. For convenience, we set $\lambda = 0.5$ for all. The parameter $k$ represents the number of nearest attributes selected for classification; a value that is too small may result in unstable classification, while a value that is too large may lead to attribute redundancy. We chose $k = 3$ for all datasets in this paper (see Appendix C.6 for its impact).

**More Experiments.** Appendix C.3 includes the aggregation studies of the $k$-nearest attributes selection. Appendix C.4 shows that label-based VR patterns are not compatible with AttrVR patterns. Appendix C.7, C.9 includes results of generating attributes with other LLMs or VLMs, and Appendix C.8 demonstrates how AttrVR handles cases when generated attributes are of low quality.

## 7 CONCLUSION

We introduce AttrVR, which extends unimodal Visual Reprogramming to VLMs (e.g., CLIP) for downstream classification tasks, specifically targeting the CLIP's inherent ability to align visual and textual information. Instead of using template-prompted labels, AttrVR optimizes through label-based attributes, making direct use of CLIP's cross-modal alignment properties. Both theoretical analysis and experimental results show that AttrVR outperforms conventional label-based VR. The visualization results, along with ablation, aggregation, and hyper-parameter studies, validate the effectiveness of AttrVR in reprogramming CLIP. The introduction of AttrVR marks an advancement in adapting VR from repurposing single-modal pre-trained models with predefined label spaces to multimodal models.

## ACKNOWLEDGEMENT

CYC, ZSY, and FL are supported by the Australian Research Council (ARC) with grant number DE240101089, and FL is also supported by ARC with grant number LP240100101, DP230101540 and the NSF&CSIRO Responsible AI program with grant number 2303037. JZQ is supported by ARC with grant number DP240101006. This research is also supported by The University of Melbourne's Research Computing Services and the Petascale Campus Initiative. We sincerely appreciate the time and dedication of the reviewers in carefully reviewing our manuscript.

## ETHICS STATEMENT

Since the method proposed in this paper is used to improve VR performance for downstream classification tasks with CLIP, there is no potential negative impact.

## REPRODUCIBILITY STATEMENT

Appendix B offers clear explanations of the theoretical results in Section 5. For reproducing the experimental results presented in Section 6, we have included a link to our anonymous downloadable source code in the abstract. Appendix A.2.2 provides details about the open-source datasets utilized in our experiments.

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

## A  APPENDIX 1: MORE TRAINING INFORMATION

### A.1  THE PROBLEM SETTING OF VR FOR CLIP

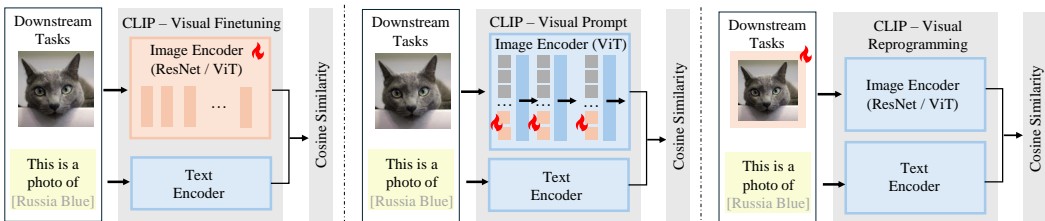

Figure 7: Different problem settings for repurposing CLIP for image classification tasks. The left shows finetuning the visual encoder, the middle illustrates a generalized approach to visual prompting, and the right depicts visual reprogramming (VR). The trainable parameters are highlighted in 'fires'. VR merely modifies the input image space, making it applicable to any encoder architecture.

The problem setting for VR and its differences from other repurposing methods for CLIP in image classification tasks are illustrated in Figure 7. For finetuning methods, the weights in the pre-trained ResNet-based or ViT-based CLIP are optimized directly using samples from the downstream task, making the Image Encoder variable. Visual prompting methods (Jia et al., 2022; Khattak et al., 2023) add parallel trainable weights next to the embedding patches in the first layer or each layer of the Image Encoder, but this approach is only applicable when ViT is used as the visual encoder.

Unlike fine-tuning or visual prompting, VR (Chen et al., 2023; Bahng et al., 2022) focuses on modifying the input space of the model rather than the pre-trained model itself. VR directly incorporates trainable parameters into the input images to achieve the repurposing of the pre-trained CLIP. Compared to other methods, VR offers the following advantages:

- Since VR only modifies the input space, it has fewer parameters (see Table 9 for details), and the number of parameters is independent of the size of the pre-trained model, only depending on the input image size. This results in *lower training time overhead*.
- By solely altering the input space, VR ensures that the original parameters of the pre-trained model remain unchanged, effectively *addressing practical issues* such as catastrophic forgetting in large models and copyright concerns.
- As the VR pattern is applied only to the images before input, it is independent of the architecture of the pre-trained model's image encoder, making it *applicable to all architectures*.
- The VR method *is orthogonal to* other fine-tuning methods—VR modifies the input space, while other methods adjust the internal parameters of the model. Therefore, VR can be combined with various methods to further enhance performance.

### A.2  IMPLEMENT DETAILS

#### A.2.1  GENERATING DESATTRS AND DISTATTRS

We used GPT-3.5 (Brown, 2020) to generate DesAttrs and DistAttrs. The specific hyper-parameter settings for text generation were as follows: temperature set to 0.99, maximum token size of 50, and generating 25 entries for each category. The termination signal was '.', and only entries with a length greater than 20 characters were considered valid and retained.

To generate DesAttrs, we queried each class in every downstream task with the following input instruction:

- `Describe the appearance of the` [Task Info.] [Class Name],

where [Task Info.] represents the description of downstream tasks, shown in Table 4, and [Class Name] represents the name of label $y^{\mathrm{T}} \in \mathcal{Y}^{\mathrm{T}}$.

To generate DistAttrs, we use the following input instruction:

- `Describe the unique appearance of a/an [Class Name] from the other [Task Info.].`

Using this approach, we successfully generated DesAttrs and DistAttrs. In the experiments, we set $m = 20$ as the size for sets $\tilde{\mathcal{A}}_{\text{des}}(y^{\text{T}})$ and $\tilde{\mathcal{A}}_{\text{dist}}(y^{\text{T}})$. When the number of valid entries generated for class $y^{\text{T}}$ is less than $m$, we randomly resampled to ensure that each class had exactly 20 attributes, facilitating subsequent experiments.

### A.2.2 DATASET DETAILS

Table 4: Dataset Information

| | Aircraft | Caltech | Cars | DTD | ESAT | Flowers | Food | Pets | SUN | UCF | IN | Resisc |
|---|---|---|---|---|---|---|---|---|---|---|---|---|
| Task Info. | aircraft model | object | fine-grained automobile | texture | remote sensing land cover | flower | food | pet | scene | action | object | remote sensing scene |
| Class Num. | 100 | 100 | 196 | 47 | 10 | 102 | 101 | 37 | 397 | 101 | 1000 | 45 |
| Batch Size | 64 | 64 | 64 | 64 | 64 | 64 | 64 | 64 | 64 | 64 | 64 | 64 |

This paper establishes benchmarks for downstream classification tasks following prior work (Oh et al., 2023), employing the same methodology to split the 16-shot training, validation, and test sets. The 12 datasets are listed as follows: FGVCAircraft (Aircraft) (Maji et al., 2013), Caltech101 (Caltech) (Fei-Fei et al., 2004), StanfordCars (Cars) (Krause et al., 2013), Texture (DTD) (Cimpoi et al., 2014), EuroSAT (ESAT) (Helber et al., 2019), Flowers102 (Flowers) (Nilsback & Zisserman, 2008), Food101 (Food) (Bossard et al., 2014), OxfordPets (Pets) (Parkhi et al., 2012), SUN397 (SUN) (Xiao et al., 2010), UCF101 (UCF) (Soomro et al., 2012), ImageNet (IN) (Deng et al., 2009), Resisc45 (Resisc) (Cheng et al., 2017). All image datasets are publicly available. Detailed task information and the batch size used for training VR are provided in Table 4.

### A.2.3 TRAINING VR PATTERNS

For all VR baseline methods compared in the paper, we adopted the following uniform training settings: an initial learning rate of 40, a momentum of 0.9 using the SGD optimizer (Harold et al., 1997), and a cosine annealing learning rate scheduler (Loshchilov & Hutter, 2016). The total number of learning epochs was set to 200. The experimental results represent the average across three seeds.

Regarding method-specific hyper-parameters, for VP (Bahng et al., 2022), we maintained consistency with the original work, using a VR noise pattern with a frame size of 30, as detailed in Table 9. For AR (Chen et al., 2023; Tsai et al., 2020), as noted by Tsao et al. (2024), the size of different VR patterns can impact the results. In this study, we conducted experiments with frame sizes of [8, 16, 32, 48] and selected 16 as the final frame size due to its optimal performance with fewer parameters. To ensure a fair comparison, our AttrVR adopted the same parameter settings as AR.

### A.3 DETAILS ABOUT ZERO-SHOT ATTRZS

For the $i$-th downstream image $x_i^{\text{T}}$ and certain label $y^{\text{T}}$, AttrZS is the zero-shot version of AttrVR where we do not train the VR noise pattern. AttrZS first resizes the sample to the required input size for the model, then calculates the similarity between the resized sample and the attribute descriptions of each class in a single pass, applying the similar equation of AttrVR:

$$\text{sim}_{\text{AttrZS}}(x_i^{\text{T}}, y^{\text{T}}) = \frac{\lambda}{k} \sum_{a \in \tilde{\mathcal{A}}_{\text{des}}^k} \text{sim}_{\text{CLIP}}(\tilde{x}_i^{\text{T}}, a) + \frac{1-\lambda}{k} \sum_{a' \in \tilde{\mathcal{A}}_{\text{dist}}^k} \text{sim}_{\text{CLIP}}(\tilde{x}_i^{\text{T}}, a'), \quad (10)$$

where $k, \lambda$ are hyper-parameters that are also used in AttrVR, $\tilde{x}_i^{\text{T}}$ is the resized image and $a, a'$ are attributes chosen from the attribute set $\tilde{\mathcal{A}}_{\text{des}}^k, \tilde{\mathcal{A}}_{\text{dist}}^k$. Then the label with largest $\text{sim}_{\text{AttrZS}}(x_i^{\text{T}}, y^{\text{T}})$ will be the prediction result for sample $x_i^{\text{T}}$.

## B APPENDIX 2: MORE THEORETICAL JUSTIFICATION

**Lemma 3** (*cf.* Lemma 1). *Let $\mathcal{A}_{\text{des}}(y) \subseteq \mathcal{A}(y)$ be the set of descriptive attributes for class $y$ as with Definition 2. Let $\Sigma_A$ and $\Sigma_L$ be the covariance matrices of the embeddings optimized with respect to $\mathcal{A}_{\text{des}}(y)$ and $y$, respectively. Then, for any class $y \in \mathcal{Y}$, we have $\text{Tr}\left(\Sigma_A\left(y\right)\right) \leq \text{Tr}\left(\Sigma_L\left(y\right)\right)$.*

*Proof.* Let $\mathcal{X}_y$ be the set of all images belonging to class $y$. We begin by defining the following: let $Z_L : \mathcal{X} \to \mathbb{R}^d$ be the label-based embedding function, let $Z_A : \mathcal{X} \to \mathbb{R}^d$ be the attribute-based embedding function, and let $Z_a : \mathcal{X} \to \mathbb{R}^d$ be the embedding function for a single attribute $a \in \mathcal{A}$.

Denote the mean embeddings, covariance matrices, and traces resulting from $Z_L$ by

$$\mu_L = \mathbb{E}_{x \in \mathcal{X}_y}\left[Z_L(x)\right],$$

$$\sigma_L = \mathbb{E}_{x \in \mathcal{X}_y}\left[(Z_L(x) - \mu_L)^\top (Z_L(x) - \mu_L)\right],$$

$$\text{Tr}(\sigma_L) = \mathbb{E}_{x \in \mathcal{X}_y}\left[\|Z_L(x) - \mu_L\|^2\right].$$

Similarly, for $Z_A$, we have

$$\mu_A = \mathbb{E}_{x \in \mathcal{X}_y}\left[Z_A(x)\right],$$

$$\Sigma_A = \mathbb{E}_{x \in \mathcal{X}_y}\left[(Z_A(x) - \mu_A)^\top (Z_A(x) - \mu_A)\right],$$

$$\text{Tr}(\sigma_A) = \mathbb{E}_{x \in \mathcal{X}_y}\left[\|Z_A(x) - \mu_A\|^2\right].$$

By Definition 2, we further express $Z_A(x)$ in terms of $Z_a(x)$:

$$Z_A(x) = \frac{1}{|\mathcal{A}_{\text{des}}(y)|} \sum_{a \in \mathcal{A}_{\text{des}}(y)} Z_a(x),$$

and accordingly, the attribute-mean is

$$\mu_A = [Z_A(x)] = \frac{1}{|\mathcal{A}_{\text{des}}(y)|} \sum_{a \in \mathcal{A}_{\text{des}}(y)} \mathbb{E}_{x \in \mathcal{X}_y}\left[Z_a(x)\right].$$

Then, the difference between embedding and mean is

$$Z_A(x) - \mu_A = \frac{1}{|\mathcal{A}_{\text{des}}(y)|} \sum_{a \in \mathcal{A}_{\text{des}}(y)} \left(Z_a(x) - \mathbb{E}_{x \in \mathcal{X}_y}\left[Z_a(x)\right]\right).$$

Jensen's inequality states that for any convex function $f$ and probability measure $p$, we have: $f(\mathbb{E}_p[X]) \leq \mathbb{E}_p[f(X)]$. Applying this to the squared norm (which is convex), we obtain:

$$\|Z_A(x) - \mu_A\|^2 = \left\| \frac{1}{|\mathcal{A}_{\text{des}}(y)|} \sum_{a \in \mathcal{A}_{\text{des}}(y)} \left(Z_a(x) - \mathbb{E}_{x \in \mathcal{X}_y}[Z_a(x)]\right) \right\|^2$$

$$\leq \frac{1}{|\mathcal{A}_{\text{des}}(y)|} \sum_{a \in \mathcal{A}_{\text{des}}(y)} \left\|\left(Z_a(x) - \mathbb{E}_{x \in \mathcal{X}_y}[Z_a(x)]\right)\right\|^2.$$

Taking expectations on both LHS and RHS leads to

$$\mathbb{E}_{x \in \mathcal{X}_y}\left[\|Z_A(x) - \mu_A\|^2\right] \leq \frac{1}{|\mathcal{A}_{\text{des}}(y)|} \sum_{a \in \mathcal{A}_{\text{des}}(y)} \mathbb{E}_{x \in \mathcal{X}_y}\left[\left\|\left(Z_a(x) - \mathbb{E}_{x \in \mathcal{X}_y}[Z_a(x)]\right)\right\|^2\right]. \quad (11)$$

We also know that for any attribute $a \in \mathcal{A}_{\text{des}}(y)$,

$$U_y(a) \geq U_y(a'), \quad \forall a' \in \mathcal{A}(y) \setminus \mathcal{A}_{\text{des}}(y),$$

where $U_y(a) = \frac{1}{|\mathcal{X}_y|} \sum_{x \in \mathcal{X}_y} f_a(x)$ is the frequency of attribute $a$ in class $y$. Define $\bar{Z}_a = \mathbb{E}_{x \in \mathcal{X}_y}[Z_a(x)]$ as the mean embedding for attribute $a$ in class $y$, we can then express the variance of attribute-based embedding as:

$$\mathbb{E}_{x \in \mathcal{X}_y}\left[\|Z_A(x) - \mu_A\|^2\right] = \mathbb{E}_{x \in \mathcal{X}_y}\left[\left\|\frac{1}{|\mathcal{A}_{\mathrm{des}}(y)|} \sum_{a \in \mathcal{A}_{\mathrm{des}}(y)} (Z_a(x) - \bar{Z}_a)\right\|^2\right]$$

$$= \frac{1}{|\mathcal{A}_{\mathrm{des}}(y)|} \sum_{a,a' \in \mathcal{A}_{\mathrm{des}}(y)} \mathbb{E}_{x \in \mathcal{X}_y}\left[(Z_a(x) - \bar{Z}_a)^\top (Z_{a'}(x) - \bar{Z}_{a'})\right].$$

By the Cauchy-Schwarz inequality, we have:

$$\mathbb{E}_{x \in \mathcal{X}_y}\left[(Z_a(x) - \bar{Z}_a)^\top (Z_{a'}(x) - \bar{Z}_{a'})\right] \le \sqrt{\mathbb{E}\left[\|Z_a(x) - \bar{Z}_a\|^2\right] \mathbb{E}\left[\|Z_{a'}(x) - \bar{Z}_{a'}\|^2\right]}. \quad (12)$$

Since we have already established the relationship between $\mathbb{E}_{x \in \mathcal{X}_y}\left[\|Z_A(x) - \mu_A\|^2\right]$ and $\mathbb{E}_{x \in \mathcal{X}_y}\left[\|Z_a(x) - \bar{Z}_a\|^2\right]$, we then proceed to prove that for any $a \in \mathcal{A}_{\mathrm{des}}(y)$,

$$\mathbb{E}_{x \in \mathcal{X}_y}\left[\|Z_a(x) - \bar{Z}_a\|^2\right] \le U_y(a)\mathbb{E}_{x \in \mathcal{X}_y}\left[\|Z_L(x) - \mu_L\|^2\right]. \quad (13)$$

By definition of $f_a(x)$, it takes value 1 if attribute $a$ is in $x$, and 0 otherwise. We express $Z_a(x)$ in terms of $Z_L(x)$ and $f_a(x)$:

$$Z_a(x) = f_a(x)Z_L(x) + (1 - f_a(x))\bar{Z}_a,$$

since $U_y(a)$ is the frequency of attribute $a$ in class $y$. Expanding the LHS, we have:

$$\mathbb{E}_{x \in \mathcal{X}_y}\left[\|Z_a(x) - \bar{Z}_a\|^2\right] = \mathbb{E}_{x \in \mathcal{X}_y}\left[\|f_a(x)Z_L(x) + (1 - f_a(x))\bar{Z}_a, -\bar{Z}_a\|^2\right]$$

$$= \mathbb{E}_{x \in \mathcal{X}_y}\left[f_a(x)^2 \|Z_L(x) - \bar{Z}_a\|^2\right]$$

$$= U_y(a)\mathbb{E}_{x \in \mathcal{X}_y}\left[\|Z_L(x) - \bar{Z}_a\|^2 \mid f_a(x) = 1\right]$$

$$\le U_y(a)\mathbb{E}_{x \in \mathcal{X}_y}\left[\|Z_L(x) - \mu_L\|^2\right],$$

where the second equality holds since $f_a(x)^2 = f_a(x)$ and $\mathbb{E}_{x \in \mathcal{X}_y}[f_a(x)] = U_y(a)$, the last inequality is justified based on a mild assumption that $\bar{Z}_a$ is closer to the class-specific mean embeddings than $\mu_L$ for a descriptive attribute.

Applying Eq. (12) and Eq. (13) to Eq. (11):

$$\mathbb{E}_{x \in \mathcal{X}_y}\left[\|Z_A(x) - \mu_A\|^2\right] = \frac{1}{|\mathcal{A}_{\mathrm{des}}(y)|} \sum_{a,a' \in \mathcal{A}_{\mathrm{des}}(y)} \mathbb{E}_{x \in \mathcal{X}_y}\left[(Z_a(x) - \bar{Z}_a)^\top (Z_{a'}(x) - \bar{Z}_{a'})\right]$$

$$\le \frac{1}{|\mathcal{A}_{\mathrm{des}}(y)|} \sum_{a,a' \in \mathcal{A}_{\mathrm{des}}(y)} \sqrt{U_y(a)U_y(a')}\mathbb{E}_{x \in \mathcal{X}_y}\left[\|Z_L(x) - \mu_L\|^2\right]$$

$$\le \frac{|\mathcal{A}_{\mathrm{des}}(y)|^2}{|\mathcal{A}_{\mathrm{des}}(y)|^2}\mathbb{E}_{x \in \mathcal{X}_y}\left[\|Z_L(x) - \mu_L\|^2\right]$$

$$= \mathbb{E}_{x \in \mathcal{X}_y}\left[\|Z_L(x) - \mu_L\|^2\right].$$

The second inequality follows that $U_y(a) \le 1$ for all attributes. By definition of $\mathrm{Tr}(\cdot)$, we have derived $\mathrm{Tr}(\Sigma_A) \le \mathrm{Tr}(\Sigma_L)$. $\qquad\square$

**Lemma 4** (*cf.* Lemma 2). *Let $\mathcal{A}_{\mathrm{dist}}(y) \subseteq \mathcal{A}(y)$ be the set of distinctive attributes for class $y$ as with Definition 3. Let $d_A(y, y')$ and $d_L(y, y')$ be $\ell^2$ distance between mean embeddings of two classes $y \ne y'$, optimized with respect to $\mathcal{A}_{\mathrm{dist}}(y)$ and $y$. Then, for any $y, y' \in \mathcal{Y}$, we have $d_A(y, y') \ge d_L(y, y')$ if $|\mathcal{A}_{\mathrm{dist}}(y)| > |\mathcal{Y}|$.*

*Proof.* Let $\mathcal{X}_y$ be the set of all images belonging to class $y$. We begin by defining the following: let $Z_L : \mathcal{X} \to \mathbb{R}^d$ be the label-based embedding function, let $Z_A : \mathcal{X} \to \mathbb{R}^d$ be the attribute-based embedding function. Let $T_L$ and $T_A$ be the text embedding functions for prompted class labels and attributes, respectively.

For any class $y \in \mathcal{Y}$, define: $\mu_L(y) = \mathbb{E}_{x \in \mathcal{X}_y}[Z_L(x)]$ and $\mu_A(y) = \mathbb{E}_{x \in \mathcal{Y}_x}[Z_A(x)]$. For any two classes $y, y' \in \mathcal{Y}$, define distances by

$$d_L(y, y') = \|\mu_L(y) - \mu_L(y')\|,$$
$$d_A(y, y') = \|\mu_A(y) - \mu_A(y')\|.$$

By Definition 3, for any $a \in \mathcal{A}_{\text{dist}}(y)$ and $y \neq y'$, $y, y' \in \mathcal{Y}$ we have:

$$\mathbb{E}_{x \in \mathcal{X}_y}[f_a(x)] > \mathbb{E}_{x \in \mathcal{X}_{y'}}[f_a(x)].$$

Then, we derive an inequality regarding the similarity:

$$\mathbb{E}_{x \in \mathcal{X}_y}[\text{sim}(Z_A(x), T_A(a))] > \mathbb{E}_{x \in \mathcal{X}_y}[\text{sim}(Z_A(x), T_A(a))], \tag{14}$$

for any $a \in \mathcal{A}_{\text{dist}}(y)$ and $y \neq y\prime$, where $Z_L(x) = \arg\max_z \text{sim}(z, T_L(y))$ and $Z_A(x) = \arg\max_z \frac{1}{\mathcal{A}_{\text{dist}}(y)} \sum_{a \in \mathcal{A}_{\text{dist}}(y)} \text{sim}(z, T_A(a))$, $\text{sim}(\cdot, \cdot)$ denotes the cosine similarity.

We then define a transformation $\phi(\cdot)$ that maps the embeddings to a space where the Euclidean distance corresponds to the dissimilarity in the original space, which is *isometric* with respect to the cosine similarity in the embedding space. In this space, each dimension corresponds to the dissimilarity with a specific attribute, preserving the similarity between $z$ and each attribute embedding, such that

$$\phi(z) = \left[\sqrt{1 - \text{sim}(z, T_A(a_1))}, \ldots, \sqrt{1 - \text{sim}(z, T_A(a_{|\mathcal{A}|}))}\right],$$

where $\mathcal{A} = \bigcup_{y \in \mathcal{Y}} A_{\text{dist}}(y)$. Then, for any two classes $y$ and $y'$, the distance between their mean embeddings:

$$\|\phi(\mu_A(y)) - \phi(\mu_A(y'))\|^2$$
$$= \sum_{i=1}^{|\mathcal{A}|} \left(\sqrt{1 - \text{sim}(\mu_A(y), T_A(a_i))} - \sqrt{1 - \text{sim}(\mu_A(y'), T_A(a_i))}\right)^2$$
$$= \sum_{a_i \in \mathcal{A}_{\text{dist}}(y)} \left(\sqrt{1 - \text{sim}(\mu_A(y), T_A(a_i))} - \sqrt{1 - \text{sim}(\mu_A(y'), T_A(a_i))}\right)^2$$
$$+ \sum_{a_j \notin \mathcal{A}_{\text{dist}}(y)} \left(\sqrt{1 - \text{sim}(\mu_A(y), T_A(a_j))} - \sqrt{1 - \text{sim}(\mu_A(y'), T_A(a_j))}\right)^2.$$

Referring to Eq. (14), we know that the inequality relationship accumulates because of the summation over all $|\mathcal{A}|$ terms.

We then define a similar transformation for the label-based method:

$$\phi_L(z) = \left[\sqrt{1 - \text{sim}(z, T_L(y_1))}, \ldots, \sqrt{1 - \text{sim}(z, T_L(y_{|\mathcal{Y}|}))}\right].$$

Accordingly, the distance between mean embeddings under label-based methods:

$$\|\phi_L(\mu_L(y)) - \phi_L(\mu_L(y'))\| = \sum_{i=1}^{|\mathcal{Y}|} \left(\sqrt{1 - \text{sim}(\mu_L(y), T_L(y_i))} - \sqrt{1 - \text{sim}(\mu_L(y'), T_L(y_i'))}\right)^2.$$

Denote $S_A \triangleq \sum_{a_i \in \mathcal{A}_{\text{dist}}(y)} \left(\sqrt{1 - \text{sim}(\mu_A(y), T_A(a_i))} - \sqrt{1 - \text{sim}(\mu_A(y'), T_A(a_i))}\right)^2$ and $S_L \triangleq \|\phi_L(\mu_L(y)) - \phi_L(\mu_L(y'))\|$, and the average contribution of each component by $\bar{S}_A = \frac{1}{|\mathcal{A}_{\text{dist}}(y)|} S_A$ and $\bar{S}_L = \frac{1}{|\mathcal{Y}|} S_L$, it is easy to check that $\bar{S}_A > \bar{S}_L$, because $\mathbb{E}_{x \in \mathcal{X}_y}[f_{a_i}(x)] > \mathbb{E}_{x \in \mathcal{X}_{y'}}[f_{a_i}(x)]$.

Then, as we assume $|\mathcal{A}_{\text{dist}}(y)| > |\mathcal{Y}|$ (this assumption is mild in common practical classification tasks, where $|\mathcal{Y}|$ often ranges from 10 to 100, whereas we can easily identify $|\mathcal{A}_{\text{dist}}(y)| > 100$ distinctive attributes for a class of objects because of the dimensionality of natural language vocabulary and sentences.), we have concluded that $S_A > S_L$, implying that

$$\sum_{a_i \in \mathcal{A}_{\text{dist}}(y)} \left( \sqrt{1 - \text{sim}(\mu_A(y), T_A(a_i))} - \sqrt{1 - \text{sim}(\mu_A(y'), T_A(a_i))} \right)^2$$

$$> \sum_{i=1}^{|\mathcal{Y}|} \left( \sqrt{1 - \text{sim}(\mu_L(y), T_L(y_i))} - \sqrt{1 - \text{sim}(\mu_L(y'), T_L(y_i'))} \right)^2 .$$

As $\sum_{a_j \notin \mathcal{A}_{\text{dist}}(y)} \left( \sqrt{1 - \text{sim}(\mu_A(y), T_A(a_j))} - \sqrt{1 - \text{sim}(\mu_A(y'), T_A(a_j))} \right)^2$ is non-negative, we arrive at $\|\phi(\mu_A(y)) - \phi(\mu_A(y'))\|^2 > \|\phi_L(\mu_L(y)) - \phi_L(\mu_L(y'))\|$.

Recall that transformations $\phi(\cdot)$ and $\phi_L(\cdot)$ are isometric with respect to the cosine similarity, we have

$$\|\phi(\mu_A(y)) - \phi(\mu_A(y'))\| = c \cdot d_A(y, y'),$$
$$\|\phi(\mu_L(y)) - \phi(\mu_L(y'))\| = c \cdot d_L(y, y'),$$

for some constant $c > 0$. Dividing both sides by $c$, we can conclude $d_A(y, y') > d_L(y, y')$. $\square$

## C  APPENDIX 3: MORE EXPERIMENTAL RESULTS

### C.1  MORE RESULTS OF DIFFERENT BACKBONES

Table 5: Accuracy comparison using RN50-based CLIP as the pre-trained model (Mean % ± Std %, ours are highlighted and the highest is in **bold**).

| RN50 | Aircraft | Caltech | Cars | DTD | ESAT | Flowers | Food | Pets | SUN | UCF | IN | Resisc | Avg. |
|---|---|---|---|---|---|---|---|---|---|---|---|---|---|
| ZS | 15.5 | 82.1 | 56.2 | 37.5 | 29.2 | 58.0 | 75.8 | 79.7 | 56.1 | 57.6 | 55.5 | 37.7 | 53.4 |
| AttrZS | 19.4 | 87.1 | **56.5** | 51.9 | 34.8 | **75.1** | **78.0** | 88.3 | **60.4** | 60.9 | **61.2** | 45.8 | 59.9 |
| VP | 16.2 | 80.1 | 44.0 | 43.4 | 59.7 | 53.6 | 65.3 | 77.2 | 48.8 | 52.0 | 49.7 | 47.7 | 53.2 |
| AR | 18.6 | 86.5 | 53.9 | 46.4 | 66.6 | 60.9 | 74.2 | 82.5 | 56.8 | 59.7 | 54.4 | **58.4** | 59.9 |
| AttrVR | **20.7** | **89.1** | 53.9 | **54.4** | **72.0** | 74.8 | 75.3 | **88.9** | 59.9 | **63.6** | 59.2 | 58.2 | **64.2** |

Table 6: Accuracy comparison using RN101-based CLIP as the pre-trained model (Mean % ± Std %, ours are highlighted and the highest is in **bold**).

| RN101 | Aircraft | Caltech | Cars | DTD | ESAT | Flowers | Food | Pets | SUN | UCF | IN | Resisc | Avg. |
|---|---|---|---|---|---|---|---|---|---|---|---|---|---|
| ZS | 17.1 | 86.0 | **63.9** | 39.0 | 28.1 | 59.7 | 79.6 | 81.9 | 56.5 | 58.4 | 58.7 | 44.4 | 56.1 |
| AttrZS | 20.8 | 90.9 | 63.0 | 51.4 | 35.1 | 75.8 | **81.2** | **89.5** | 62.1 | 63.7 | **63.8** | 51.2 | 62.4 |
| VP | 19.3 | 83.0 | 53.7 | 43.4 | 62.8 | 57.2 | 71.2 | 80.2 | 53.5 | 54.2 | 53.1 | 54.0 | 57.1 |
| AR | 19.5 | 89.7 | 62.0 | 46.3 | **70.4** | 60.4 | 78.0 | 84.4 | 58.4 | 60.6 | 57.9 | 60.2 | 62.3 |
| AttrVR | **23.3** | **92.0** | 62.2 | **55.6** | 70.3 | **76.2** | 79.5 | 89.3 | 62.1 | **64.5** | 62.2 | **64.5** | **66.8** |

Table 7: Accuracy comparison using ViT-B32-based CLIP as the pre-trained model (Mean % ± Std %, ours are highlighted and the highest is in **bold**).

| ViT-B32 | Aircraft | Caltech | Cars | DTD | ESAT | Flowers | Food | Pets | SUN | UCF | IN | Resisc | Avg. |
|---|---|---|---|---|---|---|---|---|---|---|---|---|---|
| ZS | 18.3 | 89.4 | **60.1** | 40.0 | 37.0 | 60.8 | 79.2 | 82.5 | 59.4 | 61.4 | 60.1 | 49.8 | 58.2 |
| AttrZS | 22.2 | 91.5 | 59.2 | 50.7 | 44.3 | 76.0 | **81.0** | 89.5 | **63.6** | 66.8 | **64.3** | 56.5 | 63.8 |
| VP | 24.3 | 92.3 | 58.6 | 54.9 | 85.9 | 71.2 | 75.0 | 86.8 | 61.0 | 67.3 | 59.0 | **73.9** | 67.5 |
| AR | 21.8 | **92.7** | 56.9 | 49.9 | 85.6 | 66.7 | 75.7 | 84.7 | 59.9 | 63.5 | 57.5 | 71.6 | 65.5 |
| AttrVR | **24.5** | 92.0 | 56.6 | **56.8** | **88.6** | **77.8** | 77.2 | **89.8** | 62.8 | **67.9** | 61.0 | **73.9** | **69.1** |

Tables 5-8 present the results of different methods using various architectures of the CLIP image encoder. For all models, we employed the same VR parameter numbers and hyper-parameter settings as those used in ViT-B16. In this configuration, the VR method requires downscaling the images

Table 8: Accuracy comparison using ViT-L14-based CLIP as the pre-trained model (Mean % ± Std %, ours are highlighted and the highest is in **bold**).

| ViT-L141 | Aircraft | Caltech | Cars | DTD | ESAT | Flowers | Food | Pets | SUN | UCF | IN | Resisc | Avg. |
|---|---|---|---|---|---|---|---|---|---|---|---|---|---|
| ZS | 29.7 | 90.4 | **77.1** | 51.1 | 55.3 | 73.7 | 88.9 | 89.0 | 65.0 | 72.9 | 71.3 | 60.4 | 68.7 |
| AttrZS | 35.3 | 94.4 | 77.0 | **61.1** | 54.4 | 85.6 | **91.4** | 94.2 | **69.8** | 75.2 | **76.2** | 63.9 | 73.2 |
| VP | 25.7 | 88.3 | 59.8 | 45.5 | 30.9 | 69.1 | 74.9 | 84.5 | 57.6 | 63.4 | 58.8 | **74.4** | 61.1 |
| AR | 31.7 | 93.0 | 75.6 | 55.9 | 70.4 | 74.7 | 89.0 | 91.5 | 65.4 | 73.9 | 69.7 | 72.2 | 71.9 |
| AttrVR | **38.2** | **96.1** | 74.8 | **61.1** | **74.9** | **85.8** | 90.0 | **94.3** | 68.6 | **77.8** | 73.8 | 70.1 | **75.5** |

and adding trainable noise at the edges or directly on the images, which can sometimes adversely affect image quality and, consequently, classification results. As a result, on certain benchmarks that demand high detail and resolution, such as Cars and Food, the accuracy may be lower than that of zero-shot learning. In summary, the following observations can be drawn from the tables:

- When comparing zero-shot methods, our proposed approach, AttrZS, which uses DesAttrs and DistAttrs in conjunction with k-nearest neighbor attribute selection, achieves an average accuracy improvement of 4.5% to 6.5% across 12 benchmarks compared to label-based zero-shot classification. This clearly demonstrates the effectiveness of using attributes instead of labels.

- When comparing VR for CLIP methods, even in cases where the baseline VR methods, such as VP and AR, exhibit underfitting with RN50 or overfitting with ViT-L14, replacing label-based VR with AttrVR consistently improves average accuracy by 3.6% to 4.5%.

- Furthermore, AttrVR outperforms AttrZS, especially in downstream tasks where there is a significant domain shift between the task domain and the CLIP pretraining domain (e.g., ESAT and Resisc). This advantage is even more pronounced when using a small image encoder (e.g., RN50), in the pre-trained CLIP model.

## C.2 MORE RESULTS OF TOP MATCHED ATTRIBUTES

Figure 8 and Figure 9 show the visualization of image samples with AttrVR patterns, and their nearest $k = 3$ DesAttrs and DistAttrs before and after training VR patterns. To illustrate the differences between individual samples within the same class, we selected two samples from the same task and class for demonstration. Figure 8 shows two samples labeled as 'banded' from the Texture task, while Figure 9 displays two samples labeled as 'Globe Thistle' from the Flowers task. From the visualization results, we can draw the following conclusions:

*The necessity of the Iterative Updating Strategy*: It is evident that the DesAttrs and DistAttrs closest to the same training sample differ before and after VR pattern training. Prior to training, the attribute descriptions that are closer to the sample often share similar keywords; for instance, the DesAttrs describing 'banded' tend to emphasize 'different bands' (Figure 8), while those describing 'Globe Thistle' focus on the height of the flowers (Figure 9). However, after VR pattern training, the DesAttrs and DistAttrs that are close to the same sample or different samples vary significantly.

*The necessity of both DesAttrs and DistAttrs*: DesAttrs and DistAttrs have different focal points. DesAttrs primarily describe the overall characteristics; for example, shown in Figure 9, in the description of Globe Thistle, keywords like 'tall' and '3 feet' would appear. In contrast, DistAttrs emphasize features that distinguish Globe Thistle from other flower categories, such as 'spherical shape' and 'globe-shaped'. During the training of the VR pattern, both the overall information of individual classes and the distinguishing features between different classes are needed. This aligns with the experimental results in Table 3.

*Combining DistAttrs with k-nearest neighbor attribute selection enhances the ability to capture unique features of individual samples*: Some distinguishing characteristics of a specific class may not be universal (i.e., some samples possess these features while others do not). In such cases, AttrVR employs k-nearest neighbor attribute selection to filter out attributes that are unique to individual samples. For instance, in the 'Banded' examples in Figure 8, the first image depicts land sediment, leading to the presence of the keyword 'rock or sediment', while the second image features diagonal stripes, resulting in the closest DistAttr including the keyword 'diagonal'.

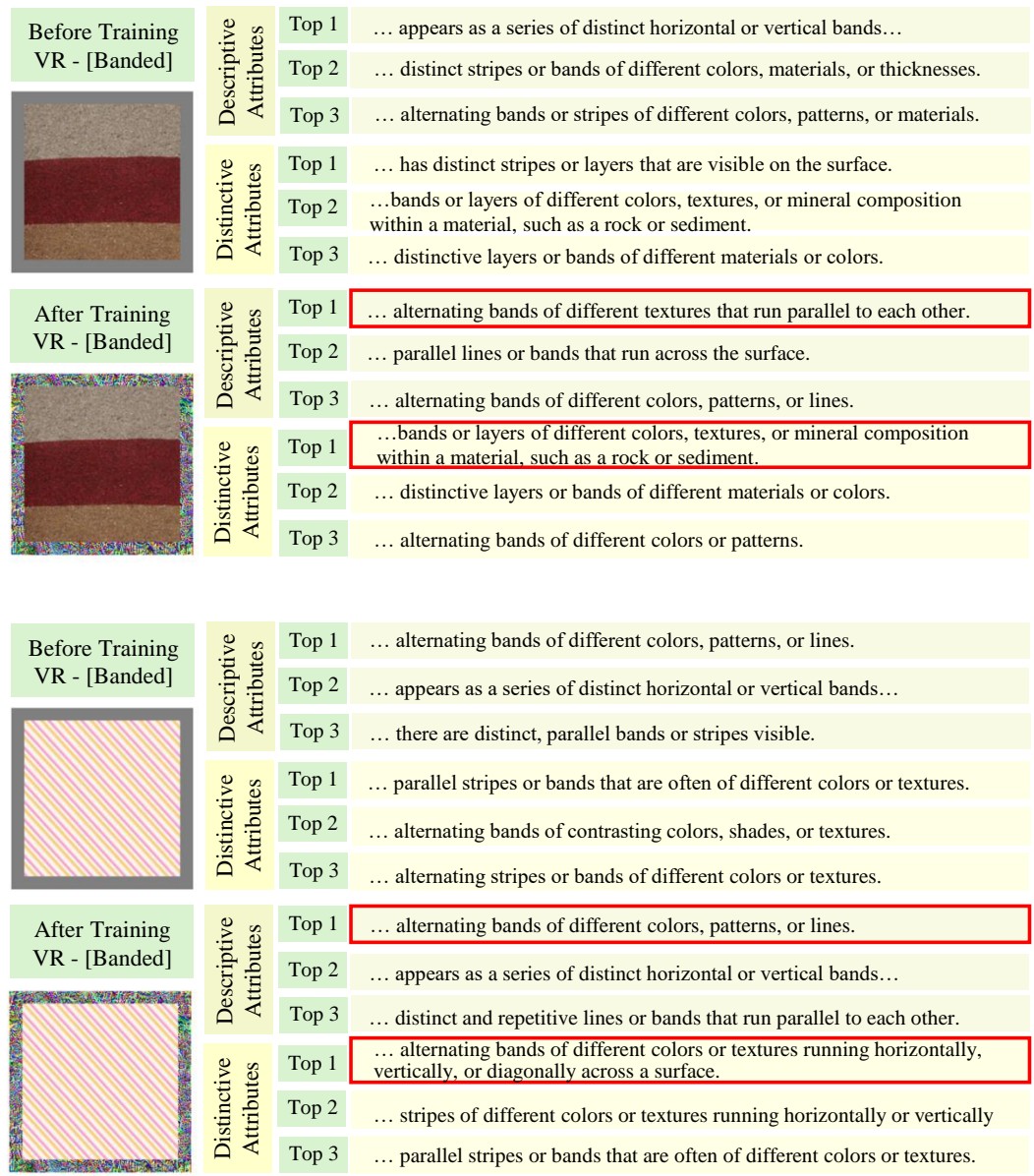

Figure 8: Visualization of images with AttrVR patterns, and their nearest $k = 3$ DesAttrs and DistAttrs before and after training VR patterns, using the ViT-B16-based CLIP as the pre-trained model. Two images labeled 'Banded' from the Texture task are chosen as examples. The closest DesAttr and DistAttr after training convergence of VR patterns are highlighted with red borders.

## C.3 MORE RESULTS OF AGGREGATION STUDIES

Aside from the $k$ nearest neighbor (*knn*) attribute selection method applied in AttrVR, we also conduct some aggregation studies replacing $f_{\mathrm{knn}}$ with the following modules to test its impact:

**The maximum similarity (*max*).** Unlike AttrVR, in this experiment, we retained only the nearest attribute with the maximum similarity in the *knn* attribute selection phase for subsequent calculations. The specific logits output $\mathrm{sim}_{\mathrm{Attr}}^{\max}(x_i^{\mathrm{T}}, y^{\mathrm{T}}|\delta^{(e)})$ given sample $x_i^{\mathrm{T}}$, label $y^{\mathrm{T}} \in \mathcal{Y}^{\mathrm{T}}$ with pattern $\delta^{(e)}$ can be expressed as:

$$\mathrm{sim}_{\mathrm{Attr}}^{\max}(x_i^{\mathrm{T}}, y^{\mathrm{T}}|\delta^{(e)}) = \lambda \mathrm{sim}_{\mathrm{CLIP}}(\tilde{x}_i^{\mathrm{T}}, a_{\mathrm{des}}|\delta^{(e)}) + (1 - \lambda)\mathrm{sim}_{\mathrm{CLIP}}(\tilde{x}_i^{\mathrm{T}}, a_{\mathrm{dist}}|\delta^{(e)}),$$

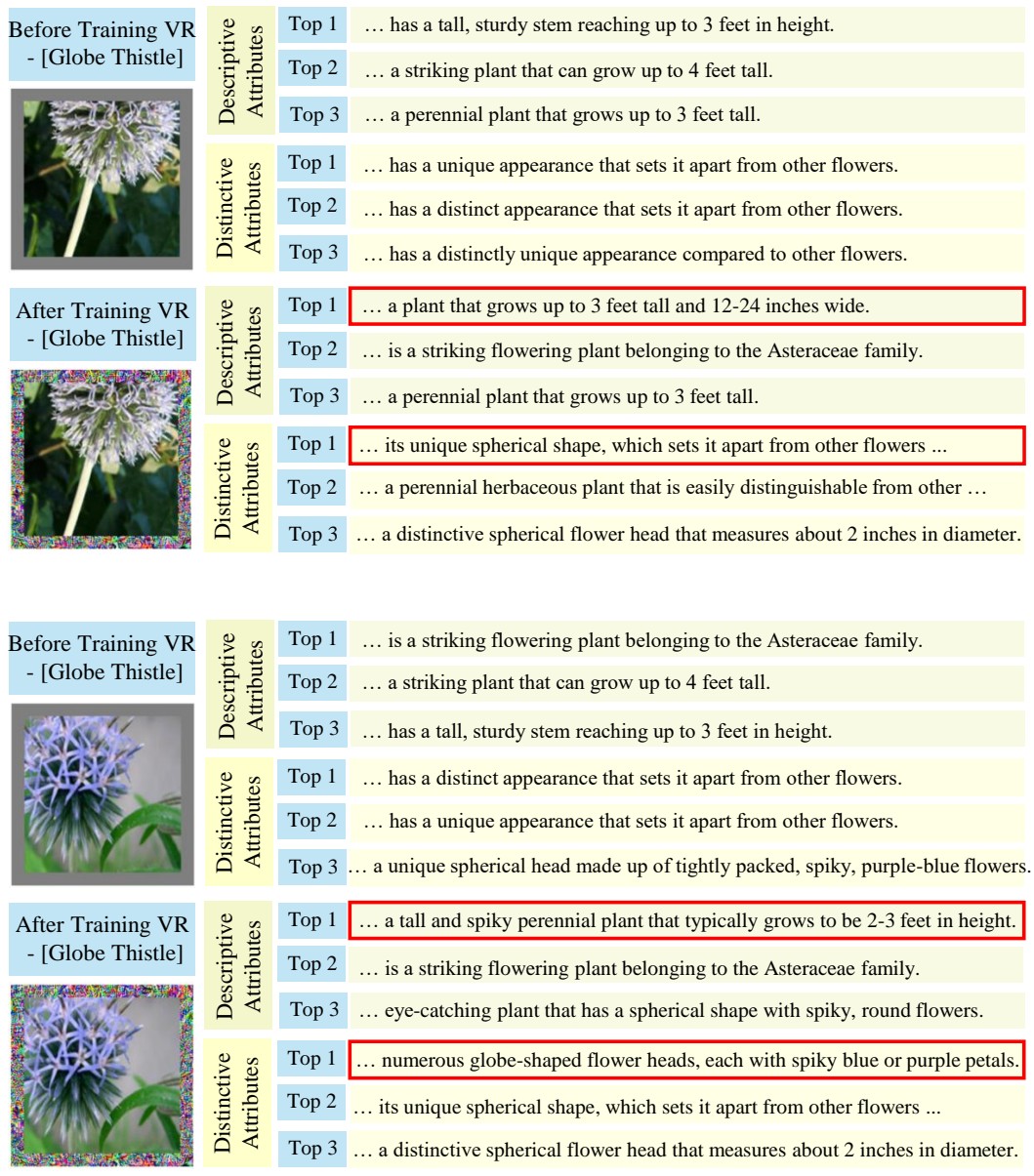

Figure 9: Visualization of images with AttrVR patterns, and their nearest $k = 3$ DesAttrs and DistAttrs before and after training VR patterns, using the ViT-B16-based CLIP as the pre-trained model. Two images labeled 'Globe Thistle' from the Flowers task are chosen as examples. The closest DesAttr and DistAttr after training convergence of VR patterns are highlighted with red borders.

where $\tilde{\mathcal{A}}_{\mathrm{des}}^{k=1}(x_i^{\mathrm{T}}, y^{\mathrm{T}}|\delta^{(e)}) = \{a_{\mathrm{des}}\}$ and $\tilde{\mathcal{A}}_{\mathrm{dist}}^{k=1}(x_i^{\mathrm{T}}, y^{\mathrm{T}}|\delta^{(e)}) = \{a_{\mathrm{dist}}\}$ can be obtained with Eq. (8) setting $k = 1$.

**The average similarity (*avg*).** In this experiment, we do not compare the similarity between various attributes and individual samples; instead, we simply calculate the average similarity between all DesAttrs, DistAttrs and a single sample. The specific logits output $\mathrm{sim}_{\mathrm{Attr}}^{\mathrm{avg}}(x_i^{\mathrm{T}}, y^{\mathrm{T}}|\delta^{(e)})$ given sample $x_i^{\mathrm{T}}$, label $y^{\mathrm{T}} \in \mathcal{Y}^{\mathrm{T}}$ with pattern $\delta^{(e)}$ can be expressed as:

$$\mathrm{sim}_{\mathrm{Attr}}^{\mathrm{avg}}(x_i^{\mathrm{T}}, y^{\mathrm{T}}|\delta^{(e)}) = \frac{\lambda}{m} \sum_{a \in \tilde{\mathcal{A}}_{\mathrm{des}}(y^{\mathrm{T}})} \mathrm{sim}_{\mathrm{CLIP}}(\tilde{x}_i^{\mathrm{T}}, a|\delta^{(e)}) + \frac{1-\lambda}{m} \sum_{a' \in \tilde{\mathcal{A}}_{\mathrm{dist}}(y^{\mathrm{T}})} \mathrm{sim}_{\mathrm{CLIP}}(\tilde{x}_i^{\mathrm{T}}, a'|\delta^{(e)}),$$

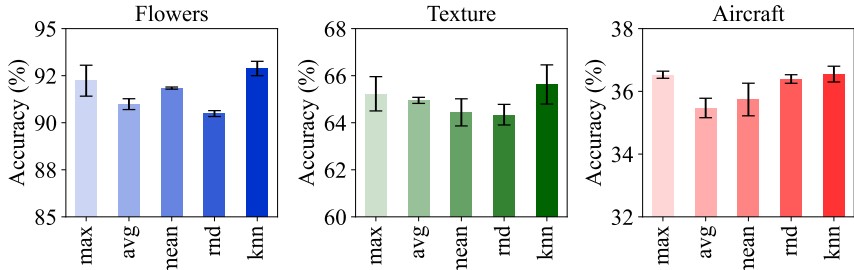

Figure 10: Results of aggregation studies, using the ViT-B16-based CLIP as the pre-trained model. '*Max*' calculates the maximum attribute similarity, '*avg*' calculates the average attribute similarity, '*mean*' calculates the similarity of the mean attribute, '*rnd*' calculates the average similarity of randomly selected $k$ attributes, and '*knn*' represents the $k$-nearest iterative updating strategy applied in AttrVR.

where $\tilde{\mathcal{A}}_{\text{des}}(y^{\text{T}})$ and $\tilde{\mathcal{A}}_{\text{dist}}(y^{\text{T}})$ are attributes set with a size of $m$, generated by Eq. (7).

**The random similarity (*rnd*).** In this experiment, we simply calculate the average similarity between $k$ randomly selected DesAttrs, DistAttrs and a single sample, where $k$ is the same hyperparameter applied in AttrVR. Similarly, the logits output $\text{sim}_{\text{Attr}}^{\text{rnd}}(x_i^{\text{T}}, y^{\text{T}}|\delta^{(e)})$ given sample $x_i^{\text{T}}$, label $y^{\text{T}} \in \mathcal{Y}^{\text{T}}$ with pattern $\delta^{(e)}$ can be expressed as:

$$\text{sim}_{\text{Attr}}^{\text{rnd}}(x_i^{\text{T}}, y^{\text{T}}|\delta^{(e)}) = \frac{\lambda}{k} \sum_{a \in \text{rand}(\tilde{\mathcal{A}}_{\text{des}}(y^{\text{T}}), k)} \text{sim}_{\text{CLIP}}(\tilde{x}_i^{\text{T}}, a|\delta^{(e)}) +$$
$$\frac{1-\lambda}{k} \sum_{a' \in \text{rand}(\tilde{\mathcal{A}}_{\text{dist}}(y^{\text{T}}), k)} \text{sim}_{\text{CLIP}}(\tilde{x}_i^{\text{T}}, a'|\delta^{(e)}), \quad (15)$$

where $\tilde{\mathcal{A}}_{\text{des}}(y^{\text{T}})$ and $\tilde{\mathcal{A}}_{\text{dist}}(y^{\text{T}})$ are attributes set with a size of $m$, generated by Eq. (7) and $\text{rand}(\mathcal{A}, k)$ is the random sampling function that chooses $k$ items from set $\mathcal{A}$.

**Mean attribute similarity (*mean*).** In this experiment, we remove the *knn* attribute selection moduel. Instead, we first compute the text embedding centers $\mathcal{Z}_{\text{des}}(y^{\text{T}})$ and $\mathcal{Z}_{\text{dist}}(y^{\text{T}})$ for the DesAttr and DistAttr sets of label $y^{\text{T}} \in \mathcal{Y}^{\text{T}}$. We then use these centers to calculate the similarity with each sample in the downstream task and update the VR pattern $\delta^{(e)}$ accordingly. The logits output $\text{sim}_{\text{Attr}}^{\text{mean}}(x_i^{\text{T}}, y^{\text{T}}|\delta^{(e)})$ of sample $x_i^{\text{T}}$ and label $y^{\text{T}}$ can be formulated as:

$$\text{sim}_{\text{Attr}}^{\text{mean}}(x_i^{\text{T}}, y^{\text{T}}|\delta^{(e)}) = \lambda \text{sim}_{\text{CLIP}}(\tilde{x}_i^{\text{T}}, \mathcal{Z}_{\text{des}}(y^{\text{T}})|\delta^{(e)}) + (1-\lambda)\text{sim}_{\text{CLIP}}(\tilde{x}_i^{\text{T}}, \mathcal{Z}_{\text{dist}}(y^{\text{T}})|\delta^{(e)}),$$
$$\text{where } \mathcal{Z}_{\text{des}}(y^{\text{T}}) = \frac{1}{m} \sum_{a \in \tilde{\mathcal{A}}_{\text{des}}(y^{\text{T}})} f_{\text{txt}}(a), \ \mathcal{Z}_{\text{dist}}(y^{\text{T}}) = \frac{1}{m} \sum_{a' \in \tilde{\mathcal{A}}_{\text{dist}}(y^{\text{T}})} f_{\text{txt}}(a').$$

**Conclusion.** The results for each module are shown in Figure 10. It can be observed that the *knn* attribute selection module in the current AttrVR achieves the highest accuracy. This is attributed to the fact that the *knn* module filters out redundant or irrelevant feature descriptions, while also determining the sample class based on the nearest attributes, thereby enhancing the robustness of classification.

## C.4 CROSS TEST BETWEEN LABEL-BASED VR AND ATTRVR

To demonstrate that the improvement in downstream task accuracy with AttrVR is due to the VR learning guided by attributes, rather than solely the attributes themselves enhancing the zero-shot accuracy during test time, we designed a cross test between the label-based VR method and AttrVR in this section.

We conduct the following experiments: (1) *Label*: Adding the VR pattern learned from labels to the images and classifying the images using the labels; (2) *Label2Attr*: Adding the VR pattern learned from labels to the images and classifying the images using the DesAttrs and DistAttrs presented in

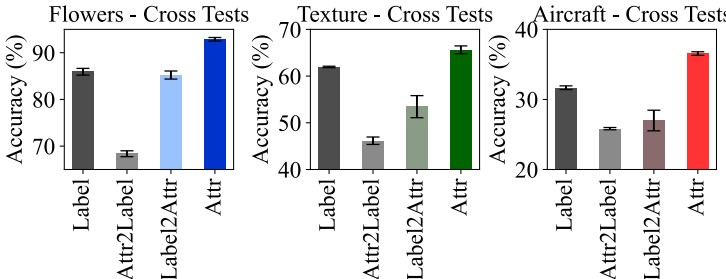

Figure 11: Cross test results between label-based VR and out AttrVR, using the ViT-B16-based CLIP as the pre-trained model. '*Label*' and '*Attr*' respectively show the results of label-based VR (i.e., AR) and AttrVR. '*Attr2Label*' shows the results of classifying downstream images with the AttrVR pattern using a template-prompted label, whereas '*Label2Attr*' shows the results of classifying downstream images with the VR pattern trained by the label-based method using DesAttrs and DistAttrs.

this paper; (3) *Attr2Label*: Adding the VR pattern learned from AttrVR to the images and classifying the images using the labels; (4) *Attr*: Adding the VR pattern learned from AttrVR to the images and classifying the images using our DesAttrs and DistAttrs.

The experimental results are shown in Figure 11. By analyzing the results, we can draw the following two conclusions:

- The performance improvement of AttrVR arises from the *VR learning process* based on attributes, rather than from the zero-shot classification performance gains obtained during testing by using attributes. It is evident that the results from *label2Attr* are significantly worse than those from AttrVR, and even slightly inferior to those from the label-based VR method. Therefore, it can be concluded that the major contribution comes from the VR learning process in AttrVR, rather than the attributes used during testing.

- The VR patterns learned through label-based VR and AttrVR are not interchangeable. The accuracy observed in the cross test (refer to *Label2Attr* and *Attr2Label* in Figure 11) is significantly lower than the accuracies obtained from both label-based VR and AttrVR. This demonstrates that the two VR patterns are not generalizable to one another. Thus, repurposing VLMs using attributes and repurposing VLMs using template-prompted labels are fundamentally different approaches. This further highlights the innovation of AttrVR.

## C.5  ANALYSIS OF TRAINING COST

### C.5.1  TIME COST COMPARISON BETWEEN BASELINES AND ATTRVR

Table 9: Training cost of different VR methods, using the ViT-B16-based CLIP as the pre-trained model and the Flowers task as an example.

|  | VP | AR | AttrVR |
|---|---|---|---|
| Parameter Number | 69.8k | 39.9k | 39.9k |
| Training Time for each Epoch (s) | 2.97±0.02 | 2.85±0.03 | 2.83±0.03 |
| Training Time in Total (min) | 9.78±0.07 | 9.44±0.05 | 9.54±0.04 |

This section provides a summary of the parameter counts and runtime for different VR methods. Experiments are conducted on a single A100 GPU.

The VP (Bahng et al., 2022) method employs a noise pattern with a frame size of 30. For an input image size of $224 \times 224$, the parameter count is calculated as $224 \times 224 \times 3 - (224 - 60) \times (224 - 60) * 3 = 69840$. In contrast, both AR (Chen et al., 2023; Tsai et al., 2020) and AttrVR (ours) use a noise pattern with a frame size of 16, resulting in a parameter count of $224 \times 224 \times 3 - (224 - 32) \times (224 - 32) * 3 = 39936$.

Detailed results are presented in Table 9. We can draw the following conclusions:

- Compared to VP and AR, the additional time incurred by our AttrVR involves calculating the text embeddings for all DesAttrs and DistAttrs once, as well as the $knn$ attributes selection once per epoch (see Algorithm 1). However, Table 9 shows that both the time for a single epoch and the total training time are similar between AttrVR and AR. Therefore, the extra time overhead introduced by AttrVR can be considered negligible.

- In comparison to VP, AR and AttrVR have fewer parameters, resulting in a lower overall training time.

### C.5.2 Time Cost of AttrVR Using Selecting Modules

Table 10: Training cost of different selecting modules of AttrVR, using the ViT-B16-based CLIP as the pre-trained model and the Flowers task as an example.

| | AttrVR (w mean) | AttrVR (w avg) | AttrVR (w max) | AttrVR (w rnd) | AttrVR (w knn) (ours) |
|---|---|---|---|---|---|
| Batch Forward Time (ms) | 7.67±0.56 | 7.57±0.43 | 7.30±0.04 | 16.63±0.10 | 7.39±0.04 |
| Training Time for each Epoch (s) | 2.88±0.08 | 2.84±0.01 | 2.86±0.04 | 2.86±0.03 | 2.83±0.03 |
| Training Time in Total (min) | 9.55±0.12 | 9.51±0.07 | 9.53±0.05 | 9.75±0.12 | 9.54±0.04 |

Table 10 shows the time for using different selection modules in AttrVR (i.e., *mean, avg, max, rnd, knn*, see Appendix C.3 for module details). It is observed that the additional computational overhead introduced by the *knn* attribute selection is negligible, though it requires sorting and averaging the nearest $k$ attributes for each sample. The reason is shown below:

Assuming there are $n$ samples and $m$ attributes for each class, the time complexity for computing the mean or maximum is $O(nm)$, while the complexity for sorting and averaging the top $k$ attributes is $O(n(m \log m + k))$. Since feature selection does not require training, the difference between these complexities is insignificant. In our case, $m = 20, k = 3$, making the computational difference almost negligible. Moreover, when compared to the computational cost of the CLIP forward pass, these overheads tend to be trivial.

### C.6 Impact of Shared and Dataset-specific Hyper-parameters

Table 11: Differences between shared and dataset-optimized hyper-parameters (using ViT-16-based CLIP as the example).

| | Aircraft | Caltech | Cars | DTD | ESAT | Flowers | Food | Pets | SUN | UCF | IN | Resisc |
|---|---|---|---|---|---|---|---|---|---|---|---|---|
| AR (Baseline) | 31.7 | 95.5 | 68.0 | 62.0 | 93.4 | 85.8 | 85.2 | 92.7 | 67.9 | 78.1 | 66.0 | 81.6 |
| AttrVR ($k$=3, $\lambda$=0.5) | 36.6 | 95.7 | 68.3 | 65.6 | 93.8 | 92.9 | 85.9 | 93.3 | 69.6 | 79.0 | 69.4 | 82.6 |
| AttrVR (optimized $k$) | 36.6 | 95.8 | 68.6 | 65.6 | 93.8 | 92.9 | 85.9 | 93.3 | 69.7 | 79.0 | 69.5 | 82.8 |
| AttrVR (optimized $\lambda$) | 37.0 | 95.9 | 68.5 | 66.0 | 93.8 | 92.9 | 85.9 | 93.3 | 70.0 | 79.0 | 69.5 | 82.6 |
| Dataset-optimized $k$ | 3 | 1 | 1 | 3 | 3 | 3 | 3 | 3 | 5 | 3 | 5 | 1 |
| Difference Between Shared and Specific $k$ | 0.0 | -0.1 | -0.3 | 0.0 | 0.0 | 0.0 | 0.0 | 0.0 | -0.1 | 0.0 | -0.1 | -0.2 |
| Dataset-optimized $\lambda$ | 0.75 | 0.25 | 0.75 | 0.25 | 0.5 | 0.5 | 0.5 | 0.5 | 0.25 | 0.5 | 0.25 | 0.5 |
| Difference Between Shared and Specific $\lambda$ | -0.4 | -0.2 | -0.2 | -0.4 | 0.0 | 0.0 | 0.0 | 0.0 | -0.4 | 0.0 | -0.1 | 0.0 |

Further experiments are conducted where we select the optimal $k$ and $\lambda$ for each dataset and compare them with shared hyper-parameters $k$ and $\lambda$. The optimized hyper-parameter values, performance, and accuracy differences are presented in Table 11. As shown, the differences between the optimal $k$ and $\lambda$ for each dataset and shared value $k = 3, \lambda = 0.5$ are minimal. Therefore, we believe that our choice for shared $k$ and $\lambda$ can be widely used across datasets.

### C.7 More Results of Using Different LLMs

It is also feasible to replace the GPT-3.5 LLM model used by our AttrVR with smaller or open-source LLMs. Even in scenarios where LLMs are unavailable, the VR training and *knn* selection

Table 12: Results of using other LLMs to generate attributes (using ViT-16-based CLIP as the example).

| LLMs | Baseline: AR | Ours: AttrVR | | | | |
|---|---|---|---|---|---|---|
| | - | Handcraft Prompts | Phi 3.1 Mini 128k | Llama 3.1 Mini | GPT-4o Mini | GPT-3.5 Turbo (ours) |
| Open-source | - | - | Yes | Yes | No | No |
| Parameters | - | - | 3.8B | 8B | NA | NA |
| Aircraft (accuracy, %) | 31.7±0.3 | 33.6±0.8 | 35.5±0.5 | 35.7±0.3 | **36.8±0.9** | 36.6±0.3 |
| DTD (accuracy, %) | 62.0±0.1 | 63.0±0.7 | 65.1±0.6 | 64.8±0.7 | **65.9±0.9** | 65.6±0.8 |
| Flowers (accuracy, %) | 85.9±0.7 | 87.5±0.6 | 89.9±0.5 | 90.1±0.1 | 92.7±0.1 | **92.9±0.4** |

modules in our AttrVR can still be utilized to obtain optimized handcrafted prompts (Radford et al., 2021) with labels for individual samples, thereby improving baseline performance. Moreover, the attribute generation process by LLMs is independent of training, which means it does not introduce additional training time overhead, and whether the LLM is open-source or not does not affect this aspect.

Table 12 shows the impact of different attribute generation methods: (1) baseline method AR (without attributes), (2) handcrafted prompts with labels (without LLMs), (3) LLM Phi 3.1 Mini 128k, (4) LLM Llama 3.1 Mini, (5) LLM GPT-4o Mini, (6) GPT-3.5 Turbo (used in AttrVR).

The following conclusions can be drawn from Table 12:

- Even when using only handcrafted prompts with labels without LLM-generated attributes, AttrVR still outperforms the baseline.

- Using smaller or open-source models can also achieve significant improvements compared to the baseline. So choosing which LLM to use might not be that important.

- Higher-quality LLMs (such as the GPT series) tend to yield slightly better results, making them more suitable for generating attributes.

## C.8 THE CASE OF GENERATED ATTRIBUTES WITH LOW QUALITY

Table 13: Results of randomly-chosen attributes and knn-chosen (ours) attributes facing attributes with different quality (using ViT-16-based CLIP as the example)

| | Normal-quality Attributes | | | | | | Low-quality Attributes | | | | High-quality Attributes | |
|---|---|---|---|---|---|---|---|---|---|---|---|---|
| | Aircraft | DTD | Flowers | Food | SUN | IN | Cars | ESAT | UCF | Resisc | Caltech | Pets |
| AR (Baseline) | 31.7 | 62.0 | 85.8 | 85.2 | 67.9 | 66.0 | 68.0 | 93.4 | 78.1 | 81.6 | 95.5 | 92.7 |
| AttrVR (w rnd) | 36.4 | 64.3 | 90.5 | 85.6 | 68.3 | 69.0 | 67.8 | 92.4 | 77.6 | 80.9 | **96.2** | **93.6** |
| AttrVR (w knn) | **36.6** | **65.6** | **92.9** | **85.9** | **69.6** | **69.4** | **68.3** | **93.8** | **79.0** | **82.6** | 95.7 | 93.3 |

Some of the generated attributes might have low qualities, and may negatively impact model performance. However, since the *knn* attribute selecting module in AttrVR is designed to select the most relevant attributes for each sample, it effectively filters out low-quality ones, ensuring the quality and relevance of the final selected attributes.

In Table 13, we present a comparison between attributes randomly selected from the generated set (marked with 'w rnd') and those selected using our *knn* module (marked with 'w knn'). When the generated attributes are of low quality, it is likely that randomly selecting attributes might yield worse results than not using attributes (e.g. Cars, ESAT, UCF, Resisc dataset). However, the *knn* module in AttrVR effectively selects high-quality and relevant attributes, leading to results that outperform the baseline method. On the contrary, when the attributes are of sufficiently high quality (e.g, Caltech, Pets), randomly selecting attributes may already achieve a high accuracy. Then the advantages of our *knn* module might diminish. In most cases, *knn* module successfully helps to ensure that only attributes with higher quality and relevance will be considered.

## C.9 MORE RESULTS OF USING MULTIMODAL LARGE LANGUAGE MODEL (MLLM) INSTEAD OF LLMS

Table 14: Results of using MLLM instead of LLMs to generate attributes (using ViT-16-based CLIP as the example).

|  | Aircraft (accuracy, %) | DTD (accuracy, %) | Flowers (accuracy, %) |
| --- | --- | --- | --- |
| AR (Baseline Method) | 31.7±0.3 | 62.0±0.1 | 85.9±0.7 |
| AttrVR (GPT-4o-mini-LLM) | **36.8±0.9** | 65.9±0.9 | 92.7±0.1 |
| AttrVR (GPT-4o-mini-VLM) | 36.6±0.4 | **68.2±0.3** | **93.7±0.4** |

The Multimodal Large Language Model (MLLM) can generate attributes and integrate seamlessly into our AttrVR framework by replacing the LLMs. In this section, the experiment is conducted using the VLM module in GPT-4o-mini to generate attributes per class based on five randomly sampled images with the size of $224 \times 224$ from the training set and the prompts below.

- DesAttr: `This is a photo of` [Class Name]. `Describe the appearance of the` [Task Info.] [Class Name].
- DistAttr: `This is a photo of` [Class Name]. `Describe the unique appearance of a/an` [Class Name] `from the other` [Task Info.].

where [Task Info.] represents the description of downstream tasks, shown in Table 4, and [Class Name] represents the name of each target class.

Comparative results with generating attributes using the LLM module in GPT-4o-mini are presented in Table 14, from which we draw the following conclusions:

- VLM-generated attributes yield better results in visually distinctive tasks (e.g., texture in DTD dataset). while their advantage over LLM-generated attributes diminishes in tasks with subtle visual differences (e.g., different models of airplanes in Aircraft dataset).
- AttrVR outperforms the baseline in all cases, using either LLM-generated or VLM-generated attributes, demonstrating its robustness regardless of the attribute generation model.

