# OpenReview forum: "Attribute-based Visual Reprogramming for Vision-Language Models"
_ICLR.cc/2025/Conference — ICLR 2025 Poster_

### Official Review · Reviewer_ELTS · 2024-11-01

**Soundness:** 4
**Presentation:** 4
**Contribution:** 3
**Rating:** 6
**Confidence:** 4

**Summary:**

The paper introduces Attribute-based Visual Reprogramming (AttrVR), a method for enhancing image classification by leveraging descriptive (DesAttrs) and distinctive attributes (DistAttrs) within vision-language models like CLIP. AttrVR replaces traditional label-based reprogramming with these attributes, refining classification through sample-specific, attribute-guided adjustments that improve intra-class consistency and inter-class separability. The approach demonstrates superior performance across multiple benchmark datasets, offering a more dynamic and attribute-focused alternative for repurposing pre-trained models in diverse downstream tasks.

**Strengths:**

1. The paper is well-written, which provides theoretical insights showing how AttrVR reduces intra-class variance and increases inter-class separation, enhancing the model’s discriminative power.
2. The motivation for this submission is easy to understand, which leverages descriptive and distinctive attributes instead of traditional label prompts, maximizing CLIP’s multimodal capabilities to improve classification accuracy.
3. AttrVR is tested across multiple datasets, with results indicating superior performance over existing methods in various visual classification tasks, especially in fine-grained categories.

**Weaknesses:**

In general, the motivation for this submission is easy to understand and insight is interesting. However, there are still several weaknesses, as follows:
1. While the paper mentions using GPT-3.5 for generating descriptive and distinctive attributes (DesAttrs and DistAttrs), it lacks detailed reasoning on why GPT-3.5 was specifically chosen over other potential models. Further, the method relies heavily on the accuracy and quality of DesAttrs and DistAttrs generated by a language model. However, there is limited discussion on how inaccuracies or noise in these attributes might affect the model's performance.
2. The effectiveness of the model appears sensitive to hyperparameters like λ (balance between DesAttrs and DistAttrs) and k (the number of nearest attributes). While the authors conduct a hyperparameter analysis, this could raise concerns regarding the robustness of the model in practical settings where optimal hyperparameters may vary widely across datasets.
3. While standard contrastive learning ablation is provided, the study lacks a comparison between hard negatives and simple negatives. Additional experiments could assess performance when hard negatives are removed, highlighting whether they significantly enhance alignment and fine-grained recognition capability.
4. Although the k-NN aggregation outperforms other methods like max, avg, and mean aggregation in attribute selection, the iterative querying process for the k-nearest attributes per sample per epoch raises concerns about computational efficiency and scalability, especially for large datasets. In addition, the sample-specific optimization approach using k-NN may cause the model to overfit to specific samples instead of capturing generalizable features.

**Questions:**

See Weaknesses

---

> ### Author Response · Authors · 2024-11-19
> **Responses to W1 (Use of Large Language Model)**
>
> ## FLexibility with Other LLMs (i.e., Smaller-Scale or Open-Source ones)
> > (W1) While the paper mentions using GPT-3.5 for generating descriptive and distinctive attributes (DesAttrs and DistAttrs), it lacks detailed reasoning on why GPT-3.5 was specifically chosen over other potential models.
>
> Thank you very much for your concern. In response to the question regarding the selection of other potential large language models (LLMs), we have added the results for additional models (Phi 3.1 Mini 128k, Llama 3.1 Mini, and GPT-4o Mini) in the table below.
>
> |                            | Aircraft (accuracy, %) | DTD (accuracy, %) | Flowers (accuracy, %) |
> |----------------------------|------------------------|-------------------|-----------------------|
> | AR (Baseline Method)       | 31.7±0.3               | 62.0±0.1          | 85.9±0.7              |
> | AttrVR (Phi 3.1 Mini 128k) | 35.5±0.5               | 65.1±0.6          | 89.9±0.5              |
> | AttrVR (Llama 3.1 Mini)    | 35.7±0.3               | 64.8±0.7          | 90.1±0.1              |
> | AttrVR (GPT-4o Mini)       | **36.8±0.9**           | **65.9±0.9**      | 92.7±0.1              |
> | AttrVR (GPT-3.5 Turbo)     | 36.6±0.3               | 65.6±0.8          | **92.9±0.4**          |
>
> **Key conclusions**
> - Less powerful LLMs also deliver competitive results. AttrVR **with other LLMs also achieves significant performance improvements** compared to the baseline. Therefore, choosing which LLM to use *might not be that important*.
> - LLMs with higher generation quality (e.g., the GPT series) **yield better results**, making the selection of such models more reasonable.
> - However, AttrVR is not tied to GPT-3.5. It can seamlessly integrate smaller or open-source LLMs, and even exclude using them to achieve tangible performance. You may also refer to [our another response](https://openreview.net/forum?id=j964C6y92q&noteId=0kk9IuxSCC) for more discussions.
>
> ---
>
> ## AttrVR is Robust to Noise/Inaccuracy in Attributes
> > (W1 cont'd) Further, the method relies heavily on the accuracy and quality of DesAttrs and DistAttrs generated by a language model. However, there is limited discussion on how inaccuracies or noise in these attributes might affect the model's performance.
>
> We confirm that the generated attributes do contain some noise and irrelevant information. However, the **$k$-nn attribute selection module** (proposed on Page 6) is specifically designed to select the most relevant attributes for each sample. This step effectively filters out low-quality attributes, ensuring the quality and relevance of the final chosen attributes. To demonstrate its effectiveness, we conducted additional experiments comparing the performance of *random attributes selection* with our *$k$-nn* attributes selection. The findings are presented in the table below.
>
> |                            | Aircraft     | Caltech101*  | Cars^        | DTD          | EuroSAT^     | Flowers      | Food101      | Pets*        | SUN397       | UCF101^      | ImageNet     | Resisc^      |
> |----------------------------|--------------|--------------|--------------|--------------|--------------|--------------|--------------|--------------|--------------|--------------|--------------|--------------|
> | AR (Baseline)              | 31.7±0.3     | 95.5±0.2     | 68.0±0.3     | 62.0±0.1     | 93.4±0.1     | 85.8±0.7     | 85.2±0.1     | 92.7±0.1     | 67.9±0.3     | 78.1±0.2     | 66.0±0.0     | 81.6±0.3     |
> | AttrVR (with random Attrs) | 36.4±0.1     | **96.2±0.1** | 67.8±0.1     | 64.3±0.4     | 92.4±0.4     | 90.5±0.2     | 85.6±0.0     | **93.6±0.1** | 68.3±0.2     | 77.6±1.0     | 69.0±0.0     | 80.9±0.6     |
> | AttrVR (with kNN Attrs)    | **36.6±0.3** | 95.7±0.1     | **68.3±0.3** | **65.6±0.8** | **93.8±0.3** | **92.9±0.4** | **85.9±0.1** | 93.3±0.0     | **69.6±0.1** | **79.0±0.6** | **69.4±0.0** | **82.6±0.4** |
>
> **Key conclusions**
> - **Low-quality attributes degrade performance**. For example, in datasets marked with **'^'** (e.g., Cars, EuroSAT, UCF101, Resisc), results from randomly selected attributes perform worse than not using attributes at all (i.e., the baseline AR).
>  - ***$k$-nn* module ensures that only high-quality attributes are used**. AttrVR with $k$-nn consistently outperforms both baseline and random selection, confirming its ability to select high-quality and relevant attributes.
>  - When the generated attributes are of sufficiently high quality, such as in datasets marked with **'\*'**, the advantage of *knn* over random attribute selection gradually diminishes.
>
> The results show that our *$k$-nn* module is critical for ensuring attribute quality and addressing concerns about reliability.
>
> We appreciate your questions and this supplementary analysis has been included in Appendix C.7 and C.8 of the revised version.

---

> ### Author Response · Authors · 2024-11-19
> **Responses to W2 (Hyperparameter Selection) and W3 (Hard Negative)**
>
> Thank you very much for your review. For each of your concerns, we first *quote* the original question, and present point-by-point clarification accordingly.
>
> ---
>
> ## AttrVR is Not Sensitive to Hyperparameter Selection
> > (W2) The effectiveness of the model appears sensitive to hyperparameters like λ (balance between DesAttrs and DistAttrs) and k (the number of nearest attributes). While the authors conduct a hyperparameter analysis, this could raise concerns regarding the robustness of the model in practical settings where optimal hyperparameters may vary widely across datasets.
>
> Thank you for your concerns. In fact, as shown in Figure 6, as long as $k$ and $\lambda$ are within a reasonable range, their fluctuations do not have a significant impact (e.g., $k \in [1, 5], \lambda \in [0.25, 0.75]$). Additionally, we have conducted further experiments where we selected the optimal $k$ and $\lambda$ for each dataset. The results are presented in the table below.
>
> |                                                    | Aircraft | Caltech101 | Cars     | DTD      | EuroSAT  | Flowers  | Food101  | Pets     | SUN397   | UCF101   | ImageNet | Resisc   |
> |----------------------------------------------------|----------|-------------|----------|----------|----------|----------|----------|----------|----------|----------|----------|----------|
> | AR (Baseline)                                      | 31.7±0.3 | 95.5±0.2    | 68.0±0.3 | 62.0±0.1 | 93.4±0.1 | 85.8±0.7 | 85.2±0.1 | 92.7±0.1 | 67.9±0.3 | 78.1±0.2 | 66.0±0.0 | 81.6±0.3 |
> | AttrVR ($k$=3, $\lambda$=0.5)                      | 36.6±0.3 | 95.7±0.1    | 68.3±0.3 | 65.6±0.8 | 93.8±0.3 | 92.9±0.4 | 85.9±0.1 | 93.3±0.0 | 69.6±0.1 | 79.0±0.6 | 69.4±0.0 | 82.6±0.4 |
> | AttrVR (optimized $k$ for each dataset)            | 36.6±0.3 | 95.8±0.2    | 68.6±0.2 | 65.6±0.8 | 93.8±0.3 | 92.9±0.4 | 85.9±0.1 | 93.3±0.0 | 69.7±0.2 | 79.0±0.6 | 69.5±0.1 | 82.8±0.5 |
> | AttrVR (optimized $\lambda$ for each   dataset)    | 37.0±0.5 | 95.9±0.0    | 68.5±0.2 | 66.0±0.3 | 93.8±0.3 | 92.9±0.4 | 85.9±0.1 | 93.3±0.0 | 70.0±0.2 | 79.0±0.6 | 69.5±0.1 | 82.6±0.4 |
> | Optimized $k$ for Specific Dataset                 | 3        | 1           | 1        | 3        | 3        | 3        | 3        | 3        | 5        | 3        | 5        | 1        |
> | Difference Between Shared and Specific   $k$       | **0.0**  | **-0.1**    | **-0.3** | **0.0**  | **0.0**  | **0.0**  | **0.0**  | **0.0**  | **-0.1** | **0.0**  | **-0.1** | **-0.2** |
> | Optimized $\lambda$ for Specific Dataset           | 0.75     | 0.25        | 0.75     | 0.25     | 0.5      | 0.5      | 0.5      | 0.5      | 0.25     | 0.5      | 0.25     | 0.5      |
> | Difference Between Shared and Specific   $\lambda$ | **-0.4** | **-0.2**    | **-0.2** | **-0.4** | **0.0**  | **0.0**  | **0.0**  | **0.0**  | **-0.4** | **0.0**  | **-0.1** | **0.0**  |
>
> **Key conclusions**
> - The differences between the optimal $k$ and $λ$ for each dataset and the results obtained with $k=3$ and $λ=0.5$ are **minimal**.
> - We believe that this choice **can be widely used across datasets**.
>
> We further note that AttrVR is under a *standard supervised learning paradigm*, where hyperparameter tuning can be a routine and manageable aspect of model training. Thus, it is easy to include any hyperparameter selection processes in AttrVR when necessary. We thus conclude that there is no further concern about hyperparameter sensitivity with our method.
>
> Thank you again for your valuable feedback, we have included this supplementary analysis in Appendix C.6 of the revision.
>
> ---
>
> ## Difficulty in Isolating Hard Negative Completely
> > (W3) While standard contrastive learning ablation is provided, the study lacks a comparison between hard negatives and simple negatives. Additional experiments could assess performance when hard negatives are removed, highlighting whether they significantly enhance alignment and fine-grained recognition capability.
>
> Thank you very much for your interesting suggestion. Below are clarification and additional context regarding hard and simple negatives and their contributions to our method.
>
> **Why is isolating hard negative difficult**?
> Identifying if a single attribute description represents **a hard or simple negative is inherently ambiguous**. For example, the sentence "The British Shorthair is a medium-sized cat with a solid, muscular build" may contain both a hard negative description ("a medium-sized cat") *shared with another class (in Fig. 1)*, and a simple negative description ("a solid, muscular build"). This overlap makes it impractical to establish an absolute categorization.
>
> **Validated Contributions via Ablation Studies**.
> Besides, the DesAttr in our method is somewhat analogous to hard negatives, while DistAttr resembles simple negatives. To assess their contributions, we have performed ablation studies where DesAttr was removed, as shown in Table 3 in the paper, which *has already* demonstrated its impact on the results.

---

> ### Author Response · Authors · 2024-11-19
> **Responses to W4 (Time Complexity of k-NN Attributes Selection against Alternative Selection Strategies)**
>
> ## Querying $k$-NN Attributes does NOT Incur Significant Time Cost
>
> > (W4) Although the k-NN aggregation outperforms other methods like max, avg, and mean aggregation in attribute selection, the iterative querying process for the k-nearest attributes per sample per epoch raises concerns about computational efficiency and scalability, especially for large datasets. In addition, the sample-specific optimization approach using k-NN may cause the model to overfit to specific samples instead of capturing generalizable features.
>
> Thank you for your concern. In fact, the additional computational overhead introduced by the *$k$-nn* attribute selection is **negligible** because
> - The attributes' text embeddings are precomputed and remain unchanged throughout training.
> - The additional overhead of the *$k$-nn* module **lies solely in the sorting and selection processes** once the similarity matrix between image samples and attributes is computed.
> - Concretely, in our $k$-NN selection, the complexity for sorting and averaging the top $k$ attributes is $O(n(m\log⁡ m+k))$, where $m$ is the number of attributes; while the time complexity for computing the *mean* or *maximum* (i.e., alternative selection strategies) is $O(nm)$.
> - For each attribute class ($m=20$ items) and top-$k=3$ selection, these operations incur minimal computational cost compared to the forward pass of CLIP, which actually constitutes the dominant time complexity in practice.
>
> To illustrate this, we measured the time required for different methods.
> The table below shows the time per batch, epoch, and total training time for AttrVR using different modules (*mean*, *avg*, *max*, and *$k$-nn*). As observed, the additional time cost of using *knn* is **indeed negligible**.
> |                           | AttrVR+mean | AttrVR+avg | AttrVR+max |**AttrVR+knn (ours)** |
> |---------------------------|-------------|------------|------------|------------|
> | Batch Forward Time (ms)   | 7.67±0.56   | 7.57±0.43  | 7.30±0.04  | 7.39±0.04             |
> | Epoch Time (s)            | 2.88±0.08   | 2.84±0.01  | 2.86±0.04  | 2.83±0.03             |
> | Whole Training Time (min) |  9.55±0.12   | 9.51±0.07  | 9.53±0.05  | 9.54±0.04             |
>
> In the revised version, we have included this explanation in Appendix C.5.
>
> ---
>
> ## $k$-NN Selection is Not Empirically Prone to Overfitting
> Additionally, regarding the potential for overfitting caused by knn, the results in Appendices C.3 and C.8 demonstrate that *$k$-nn* has an advantage in selecting and filtering attributes. The results show that it outperforms other methods, and the performance improvement far outweighs the potential impact of overfitting.
>
> Once again, **thank you for your valuable suggestions**, which have contributed to improving the quality of our paper. We hope that our reply addresses your concerns.

---

> ### Comment · Reviewer_ELTS · 2024-11-22
> **Response to author’s rebuttal**
>
> After reviewing the authors' rebuttal, I find my concerns  are almost addressed. Given the responses, I recommend maintaining the original score.

---

> ### Author Response · Authors · 2024-11-22
> **Glad to hear your concerns are almost addressed!**
>
> Dear Reviewer ELTS,
>
> Many thanks for your reply! Since there are still five days until the end of the discussion, may we know your concerns if any (as your concerns are "almost addressed" instead of "fully addressed")?
>
> We are not sure if there is anything we can do to make our paper even better? We can further strengthen our paper based on your new comments until our paper is good to be accepted. ^^
>
> Looking forward to hearing back from you.
>
> Best,
>
> The authors of Submission 6637

---

### Official Review · Reviewer_UhVi · 2024-11-01

**Soundness:** 4
**Presentation:** 3
**Contribution:** 3
**Rating:** 6
**Confidence:** 4

**Summary:**

This paper proposes an adaptation technique to improve pre-trained vision-language models for downstream classification tasks. The method uses visual reprogramming, a technique that alters the space of the input to the model – that is a learnt padding. This is done by leveraging the vast knowledge of LLMs to generate descriptive details, thus allowing to harness the potential of the joint embedding training of vision-language models, unlike prior works that only employ the class label used in the cross entropy objective. Distinctive attributes and descriptive attributes of classes together with a few clever training time tricks improves the downstream classification accuracy of VLMs.

**Strengths:**

- The paper presents a neat incorporation of the vast knowledge of LLMs to improve downstream classification of vision language models. The use of descriptive and distinctive attributes of classes to leverage the language capability in vision-language joint pre-training is well validated through experimental results and theoretical backing.
- The proposed method achieves excellent improvements above existing visual reprogramming methods across a diverse set of visual classification datasets and different CLIP backbones.

**Weaknesses:**

- Is there a way to ascertain that the distinctive captions generated are actually distinct in the embedding space? While lines 78-79 claims this is the case in Figure 1, I disagree. The distinctive attributes seem to not necessarily be farther from the cluster center compared to descriptive attributes. The authors could compare of cosine distance of descriptive attributes and distinctive attributes with the class labels. Ideally, there should be a higher distance from other classes for distinctive attributes than descriptive attributes.
- It would be interesting to see an ablation on the use of kNN for the attributes. How does it compare to randomly sampling k attributes.
- Line 167 wrong citation. I believe the intended citation is [1]. A kind suggestion for authors to double-check all their citations for accuracy

[1] Cai, Chengyi, et al. "Sample-specific Masks for Visual Reprogramming-based Prompting." arXiv preprint arXiv:2406.03150 (2024).

**Questions:**

- According to the equation for Class Separability (CS), CS would be maximized when the intra-class variance is maximized, and the inter-class distance is minimized. Can the authors clarify if this is an error in the equation (requiring a negative sign) or if there's a reason for defining CS this way that isn't apparent to me?
- AttrZS in Table 1 is unclear? How are the attributes used for zero-shot classification? The authors can add a brief explanation of this for clarity.
- Tables 5 and 6 incorrectly mention the backbone as ViT-RN. I recommend the authors carefully review all table headers and labels for consistency and accuracy throughout the paper.

---

> ### Author Response · Authors · 2024-11-19
> **Responses to Reviewer UhVi**
>
> Thank you very much for your review. For each of your concerns, we first *quote* the original question, and present point-by-point clarification accordingly.
>
> ---
>
> > (W1) Is there a way to ascertain that the distinctive captions generated are actually distinct in the embedding space? While lines 78-79 claims this is the case in Figure 1, I disagree. The distinctive attributes seem to not necessarily be farther from the cluster center compared to descriptive attributes. The authors could compare of cosine distance of descriptive attributes and distinctive attributes with the class labels. Ideally, there should be a higher distance from other classes for distinctive attributes than descriptive attributes.
>
> Thank you very much for your great suggestion. Now we have calculated the average cosine distance (i.e., $1-$ cosine similarity) between the prompted label embeddings and the *distinctive attribute* (DistAttr), *descriptive attribute* (DesAttr) embeddings for 100 attribute samples of the two classes in Figure 1. The results with mean and std are shown in the table below.
>
> |                | British Shorthair | Russia Blue |
> |----------------|-------------------|-------------|
> | DesAttr to Label  | 0.094±0.034       | 0.091±0.034 |
> | DistAttr to Label | 0.102±0.017       | 0.107±0.034 |
>
> The results confirm that DistAttr is **indeed farther** from the label embeddings compared to DesAttr. Thank you again for your valuable suggestion. We have incorporated two bar charts in Figure 1 of the revised edition to visually display the results of this comparison.
>
> > (W2) How does $k$-nn attributes selection perform compared to random attributes selection?
>
> Thank you very much for your valuable suggestion. Following your request, we **conducted additional experiments and obtained new findings**, highlighting the importance of our $k$-nn module when the quality of the generated attribute descriptions is low.
>
> |                            | Aircraft     | Caltech101*  | Cars^        | DTD          | EuroSAT^     | Flowers      | Food101      | Pets*        | SUN397       | UCF101^      | ImageNet     | Resisc^      |
> |----------------------------|--------------|--------------|--------------|--------------|--------------|--------------|--------------|--------------|--------------|--------------|--------------|--------------|
> | AR (Baseline)              | 31.7±0.3     | 95.5±0.2     | 68.0±0.3     | 62.0±0.1     | 93.4±0.1     | 85.8±0.7     | 85.2±0.1     | 92.7±0.1     | 67.9±0.3     | 78.1±0.2     | 66.0±0.0     | 81.6±0.3     |
> | AttrVR (with random Attrs) | 36.4±0.1     | **96.2±0.1** | 67.8±0.1     | 64.3±0.4     | 92.4±0.4     | 90.5±0.2     | 85.6±0.0     | **93.6±0.1** | 68.3±0.2     | 77.6±1.0     | 69.0±0.0     | 80.9±0.6     |
> | AttrVR (with kNN Attrs)    | **36.6±0.3** | 95.7±0.1     | **68.3±0.3** | **65.6±0.8** | **93.8±0.3** | **92.9±0.4** | **85.9±0.1** | 93.3±0.0     | **69.6±0.1** | **79.0±0.6** | **69.4±0.0** | **82.6±0.4** |
>
> **Key conclusions**
> - **Low-quality attributes degrade performance**. For example, in datasets marked with '^' (e.g., Cars, EuroSAT, UCF101, Resisc), results from randomly selected attributes perform worse than not using attributes at all (i.e., the baseline AR).
> - **$k$-nn module ensures that only high-quality attributes are used**. AttrVR with $k$-nn consistently outperforms both baseline and random selection, confirming its ability to select high-quality and relevant attributes.
> When the generated attributes are of sufficiently high quality, such as in datasets marked with '*', the advantage of $k$-nn over random attribute selection gradually diminishes.
>
> The results show that our $k$-nn module is critical for ensuring attribute quality and addressing concerns about reliability.
> This supplementary analysis has been included in Appendix C.8 and we also update the plot in Appendix C.3 of the revision.
>
> ---
>
> > (W3)&(Q1)&(Q3) Typos
>
> Thank you very much for your corrections. In the revised version, we have updated the relevant references (**W3**) and fixed the typo (**Q3**). Additionally, the error of the plus/minus signs in the definition formula for Class Separability (CS) (**Q1**) has been corrected. The reviewer is correct, CS should be maximized when the intra-class variance is minimized and the inter-class distance is maximized.
>
> > (Q2) Clarification of AttrZS in Table 1
>
> Thank you very much for raising this concern. Here, AttrZS refers to the scenario where we do not train the VR noise pattern. For a single sample, AttrZS first resizes the sample to the required input size for the model, then calculates the similarity between the sample and the attribute descriptions of each class using our AttrVR method in a single pass. The details of this process have been updated in Appendix A.3 of the revised edition.
>
> Once again, **thank you for your valuable suggestions**, which have contributed to improving the quality of our paper. We hope that our reply addresses your concerns.

---

> > ### Comment · Reviewer_UhVi · 2024-11-22
> > **Acknowledgement of reveiews**
> >
> > Thank you for the response. I have read the rebuttal response completely and my concerns have been addressed. I will keep my score unchanged.

---

> > > ### Author Response · Authors · 2024-11-22
> > > **Glad to hear all of your concerns are addressed!**
> > >
> > > Dear Reviewer UhVi,
> > >
> > > Many thanks for your reply! Since there are still five days until the end of the discussion, may we know your concerns if any?
> > >
> > > Since the current recommendation is borderline instead of "a good paper", we are not sure if there is anything we can do to make our paper even better? We can further strengthen our paper based on your new comments until our paper is good to be accepted. ^^
> > >
> > > Looking forward to hearing back from you.
> > >
> > > Best,
> > >
> > > The authors of Submission 6637

---

### Official Review · Reviewer_uGif · 2024-11-08

**Soundness:** 3
**Presentation:** 3
**Contribution:** 2
**Rating:** 6
**Confidence:** 4

**Summary:**

This paper introduces a new method called Attribute-based Visual Reprogramming (AttrVR) for CLIP-based image classification tasks. Unlike traditional label-based visual reprogramming methods, AttrVR fully leverages CLIP's rich attributes and diverse textual representation capabilities. The method introduces Descriptive Attributes (DesAttrs) and Distinctive Attributes (DistAttrs), which represent the common and unique features of different categories, respectively. Additionally, AttrVR achieves more dynamic and sample-specific optimization by iteratively optimizing the $k$ nearest DesAttrs and DistAttrs for each image sample. Theoretical analysis shows that AttrVR can reduce intra-class variance and increase inter-class distance, thereby improving classification performance. Experimental results demonstrate that, whether using ViT-based or ResNet-based CLIP, AttrVR outperforms other VR methods across 12 downstream tasks. Visualization results further confirm the effectiveness of AttrVR in enhancing image-attribute alignment. In summary, AttrVR not only improves CLIP's performance in downstream image classification tasks but also provides a more effective integration scheme for transitioning visual reprogramming techniques from unimodal visual models to multimodal visual-language models.

**Strengths:**

This paper addresses the limitations of Visual Reprogramming (VR) in vision-language models like CLIP by introducing Attribute-based Visual Reprogramming (AttrVR). Unlike traditional VR methods that rely on fixed text templates and category labels, AttrVR leverages Descriptive Attributes (DesAttrs) and Distinctive Attributes (DistAttrs) generated by large language models (e.g., GPT-3.5) to capture both common and unique features of each category. During training, AttrVR dynamically selects relevant attribute descriptions for each image, optimizing input noise patterns to better capture detailed features and reduce category confusion. This attribute-based approach significantly improves classification performance across 12 downstream tasks with ViT and ResNet backbones, even with limited training samples, demonstrating robustness and effectiveness. Additionally, theoretical analysis and visualizations show that AttrVR reduces intra-class variance and enhances inter-class separability, thereby enriching the model's embedding space and offering new insights for advancing vision-language model applications. Overall, AttrVR effectively combines the strengths of visual and language models, addressing the shortcomings of label-based VR methods and fully utilizing CLIP's capability to understand rich textual descriptions, thereby advancing the application of vision-language models in downstream tasks.

**Weaknesses:**

1. **Dependence on Large Language Models:** The AttrVR method relies on large language models (e.g., GPT-3.5) to generate descriptive and distinctive attributes. This dependence may increase computational costs and limit the method's applicability in resource-constrained environments.

2. **Reliability of Attribute Generation:** The attribute descriptions generated by large language models may have issues with accuracy and relevance. If the generated attributes do not match the target categories, they could negatively impact the model's performance.

3. **Computational Complexity and Efficiency:** AttrVR queries 𝑘 nearest neighbor attributes for each image sample in every training cycle, increasing computational overhead. This may impose pressure on training time and resource requirements for large datasets.

4. **Scope of Experimental Validation:** The paper primarily conducts experiments on 12 downstream tasks. It remains unclear whether the method's effectiveness can be validated on larger or more challenging datasets.

5. **Impact of Hyperparameter Selection:** Hyperparameters in AttrVR (e.g., the number 𝑘 of nearest neighbor attributes) significantly affect model performance. While the paper selects the optimal 𝑘=3, it does not demonstrate whether this choice is equally suitable in more general scenarios across different datasets, failing to prove the method's reliability in various practical environments.

**Questions:**

1. How significantly would limited or unavailable access to these large language models impact the method's effectiveness?

2. Is it possible to use smaller or open-source language models to generate attribute descriptions to reduce dependence on large models?

3. Does the paper provide mechanisms to assess and ensure the quality and relevance of attribute descriptions generated by large language models?

4. If the generated top attributes do not match the target categories, how does it impact model performance?

5. Has consideration been given to validating the method's effectiveness on larger or more challenging datasets to ensure its generalizability?

6. Although the AttrVR method leverages learnable noise, how does its performance improve in more complex image scenarios?

---

> ### Author Response · Authors · 2024-11-19
> **Responses to W1 & Q1 & Q2 (Use of Large Langue Models)**
>
> Thank you very much for your question regarding AttrVR's reliance on large language models (LLMs) and the *feasibility of alternatives*.
> For each of your concerns, we first *quote* the question and provide clarifications accordingly.
>
> ---
>
> ## FLexibility with Smaller or Open-Source LLMs
> > (W1) Dependence on Large Language Models: The AttrVR method relies on large language models (e.g., GPT-3.5) to generate descriptive and distinctive attributes. This dependence may increase computational costs and limit the method's applicability in resource-constrained environments.
>
> >(Q2) Is it possible to use smaller or open-source language models to generate attribute descriptions to reduce dependence on large models?
>
> AttrVR is **not restricted** to GPT-3.5.
> It can seamlessly integrate **smaller or open-source LLMs** while maintaining competitive performance.
> To validate this flexibility, we conducted supplementary experiments using *GPT-4o Mini*, *Phi 3.1 Mini 128k*, and *Llama 3.1 Mini* which can still deliver strong results while substantially lowering computational demands. See the Table below for detailed results.
>
>
> ## Effectiveness of AttrVR Without LLMs
> > (Q1) How significantly would limited or unavailable access to these large language models impact the method's effectiveness?
>
> Even in scenarios where **LLMs are unavailable**, AttrVR remains effective due to
> - Handcrafted Prompts with Labels: The VR training and *$k$-nn* attributes selection modules **can still be utilized** to obtain optimized handcrafted prompts with labels for individual samples, enhancing performances over the baseline.
> - No Training Overhead: Note that attribute generation is **independent of the training process**. Thus, replacing GPT-3.5 with smaller or open-source models—or even excluding LLMs—does not increase training time.
>
> ## Supplementary Experimental Results
> To further demonstrate the flexibility and usability of AttrVR when used with other LLM options, we evaluated 6 configurations on multiple datasets:
> (1) the baseline method AR (from this paper), (2) handcrafted prompts with labels (without relying on LLMs),(3) LLM Phi 3.1 Mini 128k, (4) LLM Llama 3.1 Mini, (5) LLM GPT-4o Mini, (6) GPT-3.5 (as used in this paper).
> |                            | Open-source | Parameters | Aircraft (accuracy, %) | DTD (accuracy, %) | Flowers (accuracy, %) |
> |----------------------------|-------------|------------|------------------------|-------------------|-----------------------|
> | AR (Baseline Method)       | -           | -          | 31.7±0.3               | 62.0±0.1          | 85.9±0.7              |
> | AttrVR (Handcraft Prompts) | -           | -          | 33.6±0.8               | 63.0±0.7          | 87.5±0.6              |
> | AttrVR (Phi 3.1 Mini 128k) | Yes         | 3.8B       | 35.5±0.5               | 65.1±0.6          | 89.9±0.5              |
> | AttrVR (Llama 3.1 Mini)    | Yes         | 8B         | 35.7±0.3               | 64.8±0.7          | 90.1±0.1              |
> | AttrVR (GPT-4o Mini)       | No          | NA         | 36.8±0.9               | 65.9±0.9          | 92.7±0.1              |
> | **AttrVR (GPT-3.5 Turbo) (ours)** | No          | NA         | 36.6±0.3               | 65.6±0.8          | 92.9±0.4              |
>
>
> **Key conclusions**
>  - Even when using only handcrafted prompts with labels without LLM-generated attributes, AttrVR still outperforms the baseline.
>  - Using smaller or open-source models can also achieve significant improvements compared to the baseline.
>  - Higher-quality LLMs (such as the GPT series) tend to yield slightly better results, making them more suitable for generating attributes.
>
> These insights confirm that AttrVR applies to diverse attribute generation strategies and computational settings. We have included this supplementary analysis in Appendix C.7 of the revised edition.
>
> Thank you again for your thoughtful questions.

---

> ### Author Response · Authors · 2024-11-19
> **Responses to W2 & Q3 & Q4 (Reliability of Attribute Generation)**
>
> Thank you for your concerns. For each of your concerns, we first *quote* the question and provide clarifications accordingly.
>
> ---
>
> ## AttrVR is Robust to Low-quality Attributes
> > (W2) Reliability of Attribute Generation: The attribute descriptions generated by large language models may have issues with accuracy and relevance. If the generated attributes do not match the target categories, they could negatively impact the model's performance.
>
> > (Q4) If the generated top attributes do not match the target categories, how does it impact model performance?
>
> Thanks for raising this concern.
> We have included additional ablation studies to investigate this possibility, and find that:
> - Low-quality generated attributes *can negatively impact* the training of VR models on CLIP (AttrVR variant using random attributes selection)
> - However, it **will not impact the performance of our AttrVR**, as the proposed **$k$-nn attribute selection module**  filters out those low-quality or irrelevant attributes.
>
> The table below substantiates this claim with detailed experimental comparisons.
>
> ## Mechanisms Ensuring the Attribute Quality
> > (Q3) Does the paper provide mechanisms to assess and ensure the quality and relevance of attribute descriptions generated by large language models?
>
> Yes, the **$k$-nn attribute selection module** (proposed on Page 6) is specifically designed to select the most relevant attributes for each sample. This step effectively filters out low-quality attributes, ensuring the quality and relevance of the final selected attributes. To demonstrate its effectiveness, we conducted the following experiments. The results are presented below.
>
> ## Supplementary Experimental Results
> We present a comparison between attributes randomly selected from the generated set and those selected using our *knn* module.
>
> |                            | Aircraft | Caltech101* | Cars^    | DTD      | EuroSAT^ | Flowers  | Food101  | Pets*    | SUN397   | UCF101^  | ImageNet | Resisc^  |
> |----------------------------|----------|-------------|----------|----------|----------|----------|----------|----------|----------|----------|----------|----------|
> | AR (Baseline)              | 31.7±0.3 | 95.5±0.2    | 68.0±0.3 | 62.0±0.1 | 93.4±0.1 | 85.8±0.7 | 85.2±0.1 | 92.7±0.1 | 67.9±0.3 | 78.1±0.2 | 66.0±0.0 | 81.6±0.3 |
> | AttrVR (with random Attrs) | 36.4±0.1 | 96.2±0.1    | 67.8±0.1 | 64.3±0.4 | 92.4±0.4 | 90.5±0.2 | 85.6±0.0 | 93.6±0.1 | 68.3±0.2 | 77.6±1.0 | 69.0±0.0 | 80.9±0.6 |
> | **AttrVR (with kNN Attrs) (ours)**   | 36.6±0.3 | 95.7±0.1    | 68.3±0.3 | 65.6±0.8 | 93.8±0.3 | 92.9±0.4 | 85.9±0.1 | 93.3±0.0 | 69.6±0.1 | 79.0±0.6 | 69.4±0.0 | 82.6±0.4 |
>
> **Key conclusions**
>
> - **Low-quality attributes degrade performance**. For example, in datasets marked with **'^'** (e.g., Cars, EuroSAT, UCF101, Resisc), results from randomly selected attributes perform worse than not using attributes at all (i.e., the baseline AR).
>  - ***$k$-nn* module ensures that only high-quality attributes are used**. AttrVR with $k$-nn consistently outperforms both baseline and random selection, confirming its ability to select high-quality and relevant attributes.
>  - When the generated attributes are of sufficiently high quality, such as in datasets marked with **'\*'**, the advantage of *knn* over random attribute selection gradually diminishes.
>
> The results show that our *$k$-nn* module is critical for ensuring attribute quality and addressing concerns about reliability. We appreciate your questions and this supplementary analysis has been included in Appendix C.8 of the revised version.

---

> ### Author Response · Authors · 2024-11-19
> **Responses to W3 (Computational Complexity and Efficiency), W4 & Q5 & Q6 (Scope of Experimental Validation)**
>
> > (W3) Computational Complexity and Efficiency: AttrVR queries 𝑘 nearest neighbor attributes for each image sample in every training cycle, increasing computational overhead. This may impose pressure on training time and resource requirements for large datasets.
>
> ## Querying $k$-NN Attributes does NOT Incur Significant Time Cost
> In fact, the additional computational overhead introduced by the *$k$-nn* attribute selection is **negligible** because
> - The attributes' text embeddings are precomputed and remain unchanged throughout training.
> - The additional overhead of the *$k$-nn* module **lies solely in the sorting and selection processes** once the similarity matrix between image samples and attributes is computed.
> - For each attribute class ($m=20$ items) and top-$k=3$ selection, these matrix operations incur minimal computational cost compared to the forward pass of CLIP, which constitutes the dominant time complexity in practice.
>
> To substantiate this, we measured the time required for the baseline method and AttrVR.
>
> ## Supplementary Experimental Results
> The table below shows the time per batch, epoch, and total training time for the baseline method and AttrVR. As observed, the additional time cost of using *$k$-nn* is **indeed negligible**.
>
> |                           | AR (Baseline Method) |**AttrVR+knn(ours)** |
> |---------------------------|----------------------|-------------|
> | Batch Forward Time (ms)   | 7.36±0.17            | 7.39±0.04             |
> | Epoch Time (s)            | 2.85±0.03            | 2.83±0.03             |
> | Whole Training Time (min) | 9.44±0.05            | 9.54±0.04             |
>
> We appreciate your concern and this supplementary analysis has been included in Appendix C.5 of the revised edition.
>
> ---
>
> ## Scope of Experiments
> > (W4) Scope of Experimental Validation: The paper primarily conducts experiments on 12 downstream tasks. It remains unclear whether the method's effectiveness can be validated on larger or more challenging datasets.
>
> > (Q5) Has consideration been given to validating the method's effectiveness on larger or more challenging datasets to ensure its generalizability?
>
> > (Q6) Although the AttrVR method leverages learnable noise, how does its performance improve in more complex image scenarios?
>
> Thank you for your feedback about the scope and generalizability of our experiments. We address these below:
> 1. Dataset Scope:
>     - Our experiments cover 12 diverse tasks, following widely used benchmarks from previous classification works [1, 2], and include datasets spanning diverse visual domains such as scenes, actions, textures, and fine-grained details.
>     - Additionally, we incorporated the results from the ImageNet dataset, a well-established benchmark containing a rich variety of categories and images.
> 2. Generalization to Challenging Scenarios:
>     - To the best of our knowledge, ImageNet may have **already constituted a sufficiently large and challenging** benchmark dataset in the field of image classification.
>     - Thus, we believe the results on these datasets *should be enough* to demonstrate that AttrVR **possesses strong generalizability**.
>
> We appreciate your questions and we hope our response meets your expectations.
>
> [1] Changdae Oh, Hyeji Hwang, Hee-young Lee, YongTaek Lim, Geunyoung Jung, Jiyoung Jung, Hosik Choi, and Kyungwoo Song. Blackvip: Black-box visual prompting for robust transfer learning. In CVPR, 2023.
>
> [2] Kaiyang Zhou, Jingkang Yang, Chen Change Loy, and Ziwei Liu. Learning to prompt for vision-language models. IJCV, 2022.

---

> ### Author Response · Authors · 2024-11-19
> **Responses to W5 (Hyperparameter Selection)**
>
> ## AttrVR is Not Sensitive to Hyperparameter Selection Across Datasets
>
> > (W5) Impact of Hyperparameter Selection: Hyperparameters in AttrVR (e.g., the number 𝑘 of nearest neighbor attributes) significantly affect model performance. While the paper selects the optimal 𝑘=3, it does not demonstrate whether this choice is equally suitable in more general scenarios across different datasets, failing to prove the method's reliability in various practical environments.
>
> Thank you for raising your concern regarding hyperparameter selection in AttrVR, specifically the choice of the number of $k$.
> To address it, we conducted a detailed analysis comparing the performance of AttrVR with a shared $k=3$ across all datasets against using dataset-specific optimized $k$.
>
> The results presented in the table below show that the performance (accuracy) differences between these two settings are **minimal**, with variations no greater than 0.3% across datasets. This may have demonstrated that $k=3$ is a **broadly applicable choice for diverse datasets**.
>
> We also note that AttrVR is under a standard supervised learning paradigm, where hyperparameter tuning can be a routine and manageable aspect of model training.
> By selecting $k=3$, we leverage its general applicability while avoiding extensive dataset-specific adjustments.
> That being said, it is easy to include any hyperparameter selection processes in AttrVR when necessary.
> Thus, we believe there is **no further concern about hyperparameter sensitivity** with our method.
>
> ## Supplementary Experimental Results
> |                                          | Aircraft | Caltech101 | Cars     | DTD      | EuroSAT  | Flowers  | Food101  | Pets     | SUN397   | UCF101   | ImageNet | Resisc   |
> |------------------------------------------|----------|-------------|----------|----------|----------|----------|----------|----------|----------|----------|----------|----------|
> | AR (Baseline)                            | 31.7±0.3 | 95.5±0.2    | 68.0±0.3 | 62.0±0.1 | 93.4±0.1 | 85.8±0.7 | 85.2±0.1 | 92.7±0.1 | 67.9±0.3 | 78.1±0.2 | 66.0±0.0 | 81.6±0.3 |
> | AttrVR (k=3)                             | 36.6±0.3 | 95.7±0.1    | 68.3±0.3 | 65.6±0.8 | 93.8±0.3 | 92.9±0.4 | 85.9±0.1 | 93.3±0.0 | 69.6±0.1 | 79.0±0.6 | 69.4±0.0 | 82.6±0.4 |
> | AttrVR (optimized k for each dataset)    | 36.6±0.3 | 95.8±0.2    | 68.6±0.2 | 65.6±0.8 | 93.8±0.3 | 92.9±0.4 | 85.9±0.1 | 93.3±0.0 | 69.7±0.2 | 79.0±0.6 | 69.5±0.1 | 82.8±0.5 |
> | Optimized k for Specific Dataset         | 3        | 1           | 1        | 3        | 3        | 3        | 3        | 3        | 5        | 3        | 5        | 1        |
> | Difference Between Shared and Specific k | 0.0      | -0.1        | -0.3     | 0.0      | 0.0      | 0.0      | 0.0      | 0.0      | -0.1     | 0.0      | -0.1     | -0.2     |
>
> We appreciate your concern and this supplementary analysis has been included in Appendix C.6 of the revised paper.

---

> ### Author Response · Authors · 2024-11-23
> **Follow up questions?**
>
> Dear Reviewer uGif,
>
> Thank you for taking the time to review our submission. We appreciate your thoughtful feedback and the opportunity to clarify our work through our responses and the revision.
>
> It has been a few days since we submitted our responses. If you have any further questions or require additional clarifications, we would be delighted to discuss and address them. Otherwise, we kindly hope you might consider revisiting your evaluation and possibly reflecting these clarifications in your final score.
>
> Thank you again for your time and effort in reviewing our work and your valuable contribution to improving the quality of our submission.
>
> Looking forward to hearing back from you.
>
> Best,
> The authors of Submission 6637

---

> ### Comment · Reviewer_uGif · 2024-11-23
> **Response to author’s rebuttal**
>
> Thank you very much for the your response to my questions and the detailed experimental evidence provided, which has resolved many of my doubts. However, I still have some questions that need further confirmation:
>
> Description Generation in the Inference Process:
>
> Does the inference process require generating descriptions for each image through an LLM? Without seeing the image, how is your prompt designed? The templates provided in the paper are relatively simple and may not generate very detailed and accurate descriptions. Then, the inference and classification are performed through AttrVR. The inference process of this method may be relatively complex.
>
> Possibility of Model Replacement:
>
> Can this process be replaced by more advanced LVLMs (Large Vision-Language Models)? These methods have excellent visual emergence capabilities.
>
> Detailed Explanation of Attribute Differences:
>
> Could you elaborate on the differences between DesAttrs and DistAttrs? The description in the paper is somewhat vague; it would be best to provide examples and summarize the explanation.

---

> ### Author Response · Authors · 2024-11-24
> **Response to follow-up question about inference mechanism**
>
> Thank you very much for your additional questions. Below, we still first *quote* your question, and then provide detailed responses along with additional experiments addressing your concerns.
>
> ---
>
> ### Description Generation in the Inference Process:
> > Does the inference process require generating descriptions for each image through an LLM?
>
> No, **AttrVR does not generate descriptions during inference**.
> Instead, attribute descriptions are precomputed for each class *before training* and stored as an attribute set.
> This eliminates the dependency on querying LLMs during inference.
>
> > Without seeing the image, how is your prompt designed? The templates provided in the paper are relatively simple and may not generate very detailed and accurate descriptions.
>
> Although the images are not used, the attribute-generation prompts leverage the `[target class name]` and `[downstream task name]` to generate *class-specific attributes* in the *precise downstream task context*, using the following two types of templates:
> - DesAttr: ```Describe the appearance of the [downstream task name] [target classes name]```
> - DistAttr: ```Describe the unique appearance of a/an [target classes names] from the other [downstream task name]```
>
> Simple as the templates are, they are **effective enough** to generate useful descriptions that enhance AttrVR's performance (as shown in Tables 1 and 3).
>
> While some descriptions may lack detail or accuracy, the **knn module in AttrVR addresses** this by: (1) filtering out vague or inaccurate attributes, and (2) selecting sample-specific attributes.
>
> >Then, the inference and classification are performed through AttrVR. The inference process of this method may be relatively complex.
>
> Thank you for raising this concern.
> However, there might exist misunderstandings to our method and we would like to clarify them further as follows.
> In fact, the inference process in AttrVR is straightforward and **avoids unnecessary complexity**. Here’s how it works:
> - All attribute descriptions are generated, and their text embeddings are computed before training and inference.
> - Thus, inference **only involves a standard forward pass as in training**, calculating similarity between image embeddings and precomputed text embeddings from the attribute set.
> - As the attributes text embeddings are precomputed, the **time cost during inference is negligible** (Table 10, "Batch Forward Time").
>
> ### Possibility of Model Replacement:
> > Can this process be replaced by more advanced LVLMs (Large Vision-Language Models)? These methods have excellent visual emergence capabilities.
>
> **Yes**, Multimodal Large Language Model (MLLM or LVLM) can generate attributes and integrate seamlessly into our AttrVR framework.
>
> To substantiate this, we conduct additional experiments using the GPT-4o-mini VLM module (known to have excellent visual emergence capabilities) to generate attributes for each class based on five randomly sampled images from the training set and the same prompts used in LLMs.
> *Still, both VLM- and LLM-generated attributes are provided before training*.
>
> The comparative results with generating attributes using the GPT-4o-mini LLM module are presented in the table below.
>
> |                          | Aircraft (accuracy, %) | DTD (accuracy, %) | Flowers (accuracy, %) |
> |:------------------------:|:----------------------:|:-----------------:|:---------------------:|
> |   AR (Baseline Method)   |        31.7±0.3        |      62.0±0.1     |        85.9±0.7       |
> | AttrVR (GPT-4o-mini-**LLM** module) |      **36.8±0.9**      |      65.9±0.9     |        92.7±0.1       |
> | AttrVR (GPT-4o-mini-**VLM** module) |        36.6±0.4        |    **68.2±0.3**   |      **93.7±0.4**     |
>
> **Our Concolusions**:
> - VLM-generated attributes yield better results in visually distinctive tasks (e.g., texture in DTD dataset). while their advantage over LLM-generated attributes diminishes in tasks with subtle visual differences (e.g., different models of airplanes in Aircraft dataset).
> - AttrVR outperforms the baseline in all cases, using **either** LLM-generated **or** VLM-generated attributes, **demonstrating its robustness regardless of the attribute generation model**.
>
> These results are included in Appendix C.9. of the latest revision.

---

> ### Author Response · Authors · 2024-11-24
> **Responses to follow-up question about explanation of attribute differences**
>
> > Could you elaborate on the differences between DesAttrs and DistAttrs? The description in the paper is somewhat vague; it would be best to provide examples and summarize the explanation.
>
> ### What distinguishes DesAttr and DistAttr
>
> **DesAttr** refers to the most **common visual features** across multiple image samples within the same class, describing the class by capturing its general characteristics. For instance, for classes 'pink primrose' and 'hard-leaved pocket orchid', DesAttr would describe their characteristics 'delicate appearance' and 'thin stems'.
>
> - Example of DesAttr for 'pink primrose'
>     - "The pink primrose flower has a *delicate and dainty appearance.*",
>     - "The flower pink primrose is a *dainty, delicate flower that grows on thin stems.*"
>
> - Example of DesAttr for 'hard-leaved pocket orchid'
>     - "The flower hard-leaved pocket orchid is a *small, delicate plant with a unique appearance.*",
>     - "The flower hard-leaved pocket orchid is *a small, delicate flower that grows on a thin stem.*"
>
>
> **DistAttr** highlights visual features *unique to a class*, distinguishing it from other classes. For class 'pink primrose', DistAttr would emphasize characteristics such as 'soft, pale pink color' and 'slender stem', which are **unique to** class 'pink primrose'.
> In contrast, for class 'hard-leaved pocket orchid', DistAttr would focus on characteristics like 'tightly clustered flower' and 'bell-shaped flower', features that are rarely observed in other classes.
>
> - Example of DistAttr for 'pink primrose'
>     - "A pink primrose typically has a *soft, pale pink color with delicate, ruffled petals.*",
>     - "A pink primrose has *a delicate and slender stem, typically ranging from 15-30 cm in height.*"
>
> - Example of DistAttr for 'hard-leaved pocket orchid'
>     - "A hard-leaved pocket orchid is *a small, tightly clustered flower that typically grows on trees or rocks.*",
>     - "Canterbury bells (Campanula medium) are highly recognizable flowers due to their *distinct, bell-shaped flowers that hang down from sturdy stems.*"
>
> In summary, their key difference is that DesAttr mainly describes the more common features **within the samples of a given class**, while DistAttr primarily describes the features that **distinguish the class from samples of other classes**.
> Together, these attributes enable AttrVR to balance the generality and distinctiveness of the generated attributes.
>
> ---
>
> Once again, we thank you for your thoughtful questions, helping us to improve the quality of our paper further. **Would these clarifications and additional experiments have addressed your concerns?** Looking forward to hearing back from you.

---

> > ### Comment · Reviewer_uGif · 2024-11-25
> > **Response to author’s rebuttal**
> >
> > Thank you very much for your response. I summarize that DesAttr represents the common and shared features of the target, while DistAttr represents the unique and distinguishing features of the target. The core of this work lies in how to construct a description dataset that possesses both common and distinctive features to enhance task performance. Such a dataset is very meaningful, and I can consider giving you a higher score.
> >
> > However, as far as I know, your method has very limited scalability, requiring retraining and data construction with LLMs for zero-shot tasks. Additionally, I am skeptical about whether the dataset you described is truly as effective as you claim and whether it is open-source. Moreover, with current large-scale visual-language models already performing well in fine-grained scene recognition, the value of this method is diminishing.

---

> ### Author Response · Authors · 2024-11-26
> **Response to follow-up concerns**
>
> Thank you for considering the possibility of increasing our score and some further concerns. We will address your concerns to clarify any potential misunderstandings, highlight the contributions of our work, and hope you will make your decision to increase the score. ^-^ We will quote your concerns and then provide the clarification.
>
> > The core of this work lies in how to construct a description dataset that possesses both common and distinctive features to enhance task performance.
>
> In fact, proposing the attributes dataset is **just one of** our contributions. Our **primary contribution** lies in introducing the AttrVR method to **Reprogramming** the vision-language model for downstream classification tasks (i.e., Reprogramming is a way to adapt pre-trained models to downstream tasks by only modifying the input and output spaces of downstream tasks). Our innovation compared to other reprogramming methods is that we **pair images with generated attributes for learning the reprogramming patterns**, rather than pairing them with prompted labels, which better aligns with the capability of vision-language model. We also leverage **Class Separability** (i.e., reducing intra-class variance and increasing inter-class separation) in Section 5 to theoretically explain the effectiveness of our method.
>
> > However, as far as I know, your method has very limited scalability, requiring retraining and data construction with LLMs for zero-shot tasks.
>
> In fact, our method has two versions:
>
> (1) **AttrZS**: This approach performs zero-shot classification using only generated attributes, **requiring no retraining** and involving a single CLIP forward pass, as detailed in Appendix A.3. It is **fully scalable**.
>
> (2) **AttrVR**: This version is designed for methods that need to train a *visual reprogramming pattern* (i.e., a method of adding a trainable noise pattern to the input image). This is the version we mainly discussed in the rebuttal and elaborated on extensively in the paper.
>
> Both versions show improvements over their respective baselines and do not introduce any additional computational overhead compared to the baselines. Therefore, the work presented in this paper **should not face limitations in scalability**.
>
>
> > Additionally, I am skeptical about whether the dataset you described is truly as effective as you claim and whether it is open-source.
>
> Please rest assured regarding the effectiveness and open-source concerns. Our method ensures reproducibility, and **the code is already provided** via an anonymous link in this paper. If the paper is accepted, we will update the camera-ready version **with a GitHub link and upload all generated attributes**, including those generated by other LLMs and VLMs during our rebuttal experiments.
>
> > Moreover, with current large-scale visual-language models already performing well in fine-grained scene recognition, the value of this method is diminishing.
>
> In this response, we would like to emphasize the significance of our method and the value of such research direction.
>
> Firstly, as shown in Table 1, even though vision-language models perform well in many cases, they **still struggle** on certain tasks, such as ESAT (a remote sensing dataset). Our method **helps to unlock better performance** for vision-language models in such tasks.
>
> Secondly, although the current development of vision-language models has led to high accuracy in many classification tasks, it is **still crucial to study their training and alignment** across different distributions and propose improvements. Such research not only aims to improve performance but also seeks to **promote the application of traditional mathematics and machine learning theories** in the era of foundation models.
>
> ---
> Do our responses solve your further concerns? Would it help you to make the decision about increasing the score? ^-^ Hope to hear from you.

---

> > ### Author Response · Authors · 2024-11-26
> > **Thanks for increasing your score to 6!**
> >
> > Dear Reviewer uGif,
> >
> > We are happy to see that you increase your score to 6. Many thanks for recognizing our contribution to the field!
> >
> > However, we do not see your confirmation that our latest responses addressed your follow-up comments. If so, we are happy to see that your concerns are addressed. If not, can you advise us of your remaining concerns? We are delighted to discuss this with you!
> >
> > Best,
> >
> > Authors of Submission 6637

---

### Author Response · Authors · 2024-12-03
**Summary after the rebuttal: Responses and Revisions (Part I)**

We appreciate the thorough feedback from all reviewers and have addressed each concern through additional experiments and clarifications. We outline our key responses and improvements to the manuscript in this summary.

**Responses to Reviewer uGif**

1. **Dependence on Large Language Models (LLMs)**
- Concerns: Can smaller or open-source LLMs be used? What happens if LLMs are unavailable?
- Response: We demonstrate that AttrVR is compatible with *smaller* (GPT-4o mini) or *open-source* (Phi 3.1 Mini 128k, and Llama 3.1 Mini) LLMs, and remains effective even *without LLMs* by utilizing handcrafted prompts with labels. We **add a section (Appendix C.7)** to include the results from these additional experiments.

2. **Reliability of Attribute Generation**
- Concerns: How is attribute quality assessed? What is the impact of low-quality attributes?
- Response: We demonstrate that low-quality attributes would negatively affect VR training but **not** AttrVR performance, due to the $k$-nn attributes selection module proposed to ensure attribute quality (on Page 6). We **add a section (Appendix C.8)** to include a comparative analysis between $k$-nn and random attribute selection, confirming its efficacy.

3. **Computational Complexity of $k$-nn**
- Concerns: would $k$-nn introduce significant computation overhead?
- Response: We have measured time per batch, epoch, and total training time, confirming that $k$-nn selection results in negligible overhead **(detailed in Appendix C.5)**.

4. **Scope of Experiments**
- Concerns: Does AttrVR generalize across datasets and handle complex image data?
- Response: We explain that our empirical studies follow commonly used benchmarks in CLIP-related literature, span 12 diverse datasets covering scenes, actions, textures, and fine-grained details, as well as ImageNet, which may have already constituted a sufficiently large and challenging benchmark dataset.

5. **Hyperparameter Sensitivity**
- Concerns: How sensitive is AttrVR to hyperparameter $k$?
- Response: We conduct additional experiments to show hyperparameter-induced performance variations are under 0.3%. We **append these results in Appendix C.6**.

6. **Inference Process**
- Concerns: Does the inference of AttrVR require generating descriptions for each image through LLM?
- Response: We **clarify** that AttrVR precomputes attribute descriptions *before* training, with inference involving a standard forwawrd pass. We **include timing details in Table 10**.

7. **Model Replacement**
- Concerns: Can Large Vision-Language Models be used to generate attributes?
- Response: We **report additional results in Appendix C.9** to show that AttrVR can seamlessly integrate multi-modal LLMs (e.g., GPT-4o-mini-VLM) and demonstrates robustness **regardless of the attribute generation model**.

8. **Follow-up Clarifications**

- We explain the definitions of DesAttr and DistAttr, showcase specific examples and conclude their differences.
- We clarify that AttrZS (zero-shot version of AttrVR) requires no retraining and uses a single CLIP forward pass, which **ensures scalability**.
- We clarify that our method ensures full reproducibility, and the code is already provided via an anonymous link.
- We clarify that AttrVR addresses the limitations in VLMs arising from domain shifts and distribution discrepancies, such as EuroSAT. We highlight that AttrVR advances VLM alignment and distribution adaptation, contributing to foundational research in improving model robustness across diverse tasks.



**Responses to Reviewer UhVi**

1. **Distinctiveness of attributes in Embedding Space**
- Concerns: Are DistAttr embeddings distinct from DesAttr?
- Response: We **calculate average cosine distances** to confirm DistAttr is farther from label embeddings than DesAttr. We **include the visualization** of these results in Figure 1.

2. **Performance of $k$-nn Attribute Selection**
- Concerns: How does $k$-nn compare to random attributes?
- Response: We **conduct experiments** (Appendix C.8) and **add comparative analysis** (Appendix C.3) to demonstrate that $k$-nn selection ensures higher quality attributes are used in optimization and prediction.

3. **Typos and Clarifications**
- We have corrected typos and explained that AttrZS (Table 1) is the zero-shot version of AttrVR where no training is performed, **in Appendix A.3**.

---

> ### Author Response · Authors · 2024-12-03
> **Summary after the rebuttal: Responses and Revisions (Part II)**
>
> **Responses to Reviewer ELTS**
>
> 1. **LLMs and Attribute Noise**
> - Concerns: Can other LLMs be used? How does noise in generated attributes affect performance?
> - Response: We demonstrate that AttrVR is compatible with *smaller* (GPT-4o mini) or *open-source* (Phi 3.1 Mini 128k, and Llama 3.1 Mini) LLMs. We **add a section (Appendix C.7)** to include the results from these additional experiments and discuss the noise in generated attributes **(Appendix C.8)**.
>
> 2. **Hyperparameter Sensitivity**:
> - Concerns: Is AttrVR sensitive to $k$ or $\lambda$?
> - Response: We conduct additional experiments to show hyperparameter-induced performance variations are under 0.4% **(added in Appendix C.6)**.
>
> 3. **Comparison of Negatives**:
> - Concern: How do hard and simple negatives compare?
> - Response: We clarify that DesAttr acts as hard negatives with Ablation studies (Table 3), while DistAttr serves as simple negatives.
>
> 4. **Time Complexity of $k$-nn**:
> - Concerns: Is $k$-nn efficient compared to alternatives?
> - Response: We **conduct time complexity analysis** and **measure the training time**, confirming $k$-nn selection results in negligible overhead. We **include the details in Appendix C.5.**.
>
> ---
> We believe these responses have comprehensively **addressed all reviewer concerns**, incorporating additional experiments and clarifications. The reviewers have not raised further issues and have **indicated scores leaning towards acceptance**.

---

### Meta-Review · Area_Chair_WbZK · 2024-12-20

**Metareview:**

Summary:
This paper introduces a new method called Attribute-based Visual Reprogramming (AttrVR) for CLIP-based image classification tasks. The experimental results on 12 downstream tasks are given.

The main strengths include 1) strong experimental performance, and 2) well-motivated idea.
The main weaknesses are 1) lack of ablation study on the setting of KNN, and 2) lack of evaluation on the Complexity of the k-nn module in AttrVR.

All reviewers recognized the strengths of leveraging descriptive and distinctive attributes to maximize CLIP’s multimodal capabilities, and the issues raised by the reviewers were well addressed by the authors in the discussion stage.
The AC agrees with the reviewers and recommends accepting this paper, but the authors should consider the reviewers' suggestions to include necessary experimental comparisons/analyses.

**Additional Comments On Reviewer Discussion:**

During the rebuttal phase, the authors provided more detailed experimental results and clarifications regarding the issues of ablation study, complexity analysis, and generalization to other LLMs/VLMs.

The reviewers are convinced by the feedback and one of them raises the score from 5 to 6, making the final ratings to be 6, 6, and 6.

---

### Decision · Program_Chairs · 2025-01-22

Accept (Poster)